# Lognormal firing rate distribution reveals prominent fluctuation–driven regime in spinal motor networks

Peter C Petersen, Rune W Berg*

Department of Neuroscience and Pharmacology, Faculty of Health and Medical Sciences, University of Copenhagen, Copenhagen, Denmark

**Abstract** When spinal circuits generate rhythmic movements it is important that the neuronal activity remains within stable bounds to avoid saturation and to preserve responsiveness. Here, we simultaneously record from hundreds of neurons in lumbar spinal circuits of turtles and establish the neuronal fraction that operates within either a 'mean-driven' or a 'fluctuation–driven' regime. Fluctuation-driven neurons have a 'supralinear' input-output curve, which enhances sensitivity, whereas the mean-driven regime reduces sensitivity. We find a rich diversity of firing rates across the neuronal population as reflected in a lognormal distribution and demonstrate that half of the neurons spend at least 50 % of the time in the 'fluctuation–driven' regime regardless of behavior. Because of the disparity in input–output properties for these two regimes, this fraction may reflect a fine trade–off between stability and sensitivity in order to maintain flexibility across behaviors.

## Introduction

Rhythmic movements, such as walking, scratching, chewing and breathing, consist of a recurrent sequence of activity, which is generated by neuronal networks primarily in the spinal cord and medulla. Although, this sequential activity is formed by collective communication among the neurons, it is unknown how the participation is shared versus divided within the population. Distinct motor tasks have been reported to be divided among dedicated microcircuits in zebrafish (*Ampatzis et al., 2014*; *Bagnall and McLean, 2014*; *Fetcho and McLean, 2010*). Nevertheless, do all neurons, which are dedicated to a particular motor activity, spike at approximately the same rate? Or do only some neurons spike at high rate, while most others spike at lower rates? An arrangement with a spectrum of different firing rates could be beneficial by adding the possibility of increasing the overall activity, for instance during uphill walking where a stronger force is needed. In this way the spinal circuit could enhance flexibility by adopting a diversity of firing rates across the population. Other networks in the central nervous system face a similar challenge of how to distribute the activity across the population in order to collectively increase the dynamic range (*Wohrer et al., 2013*). In sensory processing, neural circuits must be able to retain sensitivity both to weak and strong input. Weak stimuli are amplified whereas strong stimuli are attenuated in order to reduce saturation. If there is too much activity, the circuit reaches saturation and therefore loses the ability to resolve differences in sensory input. Furthermore, amplification of weak signals by recurrent excitation pose the risk of unstable activity, which can spin out of control (*Vogels et al., 2005*). This computational challenge of how networks maintain both stability and sensitivity is an open question especially for spinal networks.

Stability has primarily been investigated in cortical networks and much evidence suggest that local excitation is carefully balanced by inhibition to assure stability and to widen the range of operation (*Galarreta and Hestrin, 1998*; *Shu et al., 2003*). It is well–established that unstable states such

*For correspondence: runeb@ sund.ku.dk

**Competing interests:** The authors declare that no competing interests exist.

**eLife digest** Where and how are rhythmic movements, such as walking, produced? Many neurons, primarily in the spinal cord, are responsible for the movements, but it is not known how the activity is distributed across this group of cells and what type of activity the neurons use. Some neurons produce regular patterns of "spiking" activity, while others produce spikes at more irregular intervals. These two types of activity have different origins and represent different states of the neural network. It is not clear whether they participate equally in a movement, or if there is a hierarchy among the neurons, such that some neurons have more influence than others.

Petersen and Berg studied neurons in the lower spines of turtles during rhythmic movements. The experiments show that during rhythmic scratching some neurons are very active while most aren't particularly active at all. This is known as a lognormal distribution and is seen in many other situations, such as the levels of income of people in a society.

Petersen and Berg also found that neurons can move between two regimes of activity, called the mean-driven and fluctuation-driven spiking regimes. During rhythmic scratching, the neurons are almost equally divided between the two regimes, and this division is also found in other types of rhythmic movement. This even division between the two regimes is likely to be important for maintaining a balance between the sensitivity and stability of the neural network. The next steps following on from this work are to reveal the mechanisms behind the two regimes and to find out what causes these differences in activity.

as epileptiform activity can easily be achieved by shifting the balance in favor of excitation, e.g. by blocking inhibition (*Dichter and Ayala, 1987*; *Bazhenov et al., 2008*). The concept of *balanced excitation (E) and inhibition (I)* (balanced networks in short) was introduced two decades ago (*Shadlen and Newsome, 1994*; *van Vreeswijk and Sompolinsky, 1996*) and has sparked numerous studies both theoretical (*Amit and Brunel, 1997*; *Ozeki et al., 2009*; *van Vreeswijk and Sompolinsky, 1998*; *Kumar et al., 2008*) as well as experimental (*Berg et al., 2007*; *Okun and Lampl, 2008*; *Higley and Contreras, 2006*; *Wehr and Zador, 2003*; *Kishore et al., 2014*). The primary purpose of theoretical models of balanced networks was initially to understand irregular spiking, which was widely observed in experiments (*Bell et al., 1995*; *Shadlen and Newsome, 1994*). Irregular spiking was puzzling because it could not be explained by random arrival of excitatory input alone, since this randomness was effectively regularized by temporal integration (*Denève and Machens, 2016*; *Softky and Koch, 1993*). Models of balanced networks not only were able to explain irregular spiking, but also revealed other interesting phenomena, such as emergent linearity (*van Vreeswijk and Sompolinsky, 1996*), multifunctionalism (*Sussillo and Abbott, 2009*; *Hennequin et al., 2014*) and self–sustained stable network activity (*Amit and Brunel, 1997*; *Hansel and Mato, 2001*; *Ikegaya et al., 2013*).

The consensus view thus became that irregular spiking results from a mean membrane potential, which is lurking just below threshold, where it is restrained by inhibition concurrent with excitation (*Shadlen and Newsome, 1998*; *Bell et al., 1995*; *Salinas and Sejnowski, 2000*), although synchrony of random excitation is sometimes needed when individual synaptic potentials are small (*Stevens and Zador, 1998*). This view was essentially predicted much earlier in random walk models (*Gerstein and Mandelbrot, 1964*). The concept of balanced E/I is now an integrated part of understanding network processing in cortex and elsewhere, but for some reason it has been forgotten in understanding spinal motor networks, with the exception of a few isolated studies (*Berg et al., 2007*; *Petersen et al., 2014*).

The balanced E/I allow a subthreshold fluctuating membrane potential, where the spikes are evoked by synaptic transients and therefore belong to the *fluctuation–driven regime* (*Kuhn et al., 2004*; *Tiesinga et al., 2000*). This is in contrast to the more traditional *mean–driven spiking* (*Figure 1*), where the mean membrane potential ($V_m$) is well above threshold and spike timing is controlled by after–hyperpolarization (*Gerstner et al., 2014*; *Renart et al., 2007*). These two regimes have contrasting manifestations (*Table 1*): The fluctuation–driven regime has a skewed/lognormal firing rate distribution whereas the mean–driven regime has regular spiking and a symmetric

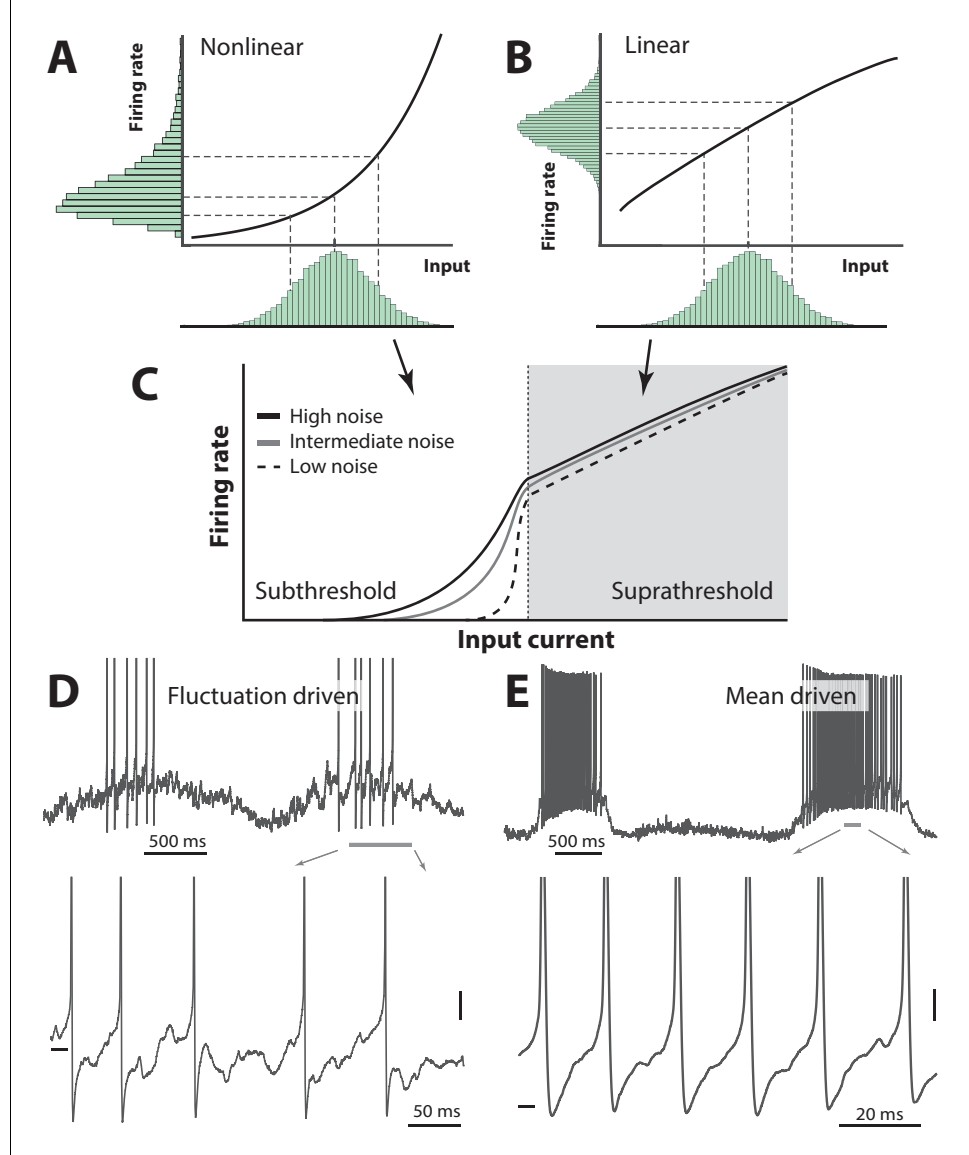

**Figure 1.** Skewness of the rate distribution reveals two regimes of neuronal spiking. (**A**) In the fluctuation–driven regime the mean input is below the spiking threshold and the IO-curve has a nonlinear shape. A normally distributed input current (shown below x–axis) is transformed into a skewed firing rate distribution (y-axis). (**B**) In contrast, if the mean input is above threshold, the transformation is linear and the firing rate distribution is symmetric. (**C**) IO–function for both regimes: Linear for suprathreshold region and nonlinear for subthreshold region. The noise level affects the curvature of the nonlinearity (3 curves illustrate different levels of noise). (**D**) Sample recordings during motor activity from two spinal neurons in the subthreshold region, where the spiking is irregular and driven by fluctuations, and the supra–threshold region (**E**), where the mean input is above threshold and spiking is regular. Highlighted area shown at bottom. Spikes in bottom panel are clipped. Tick marks: −50 mV, scale bars: 5 mV. (**A–B**) adapted from (*Roxin et al., 2011*).

distribution. A simple mechanism has been proposed to explain the lognormal firing in the fluctuation–driven regime by *Roxin et al. (2011)*: The skewness in distribution arises out of a supralinear transformation of the synaptic input, which is Gaussian by virtue of the central limit theorem (*Figure 1A*). A response to multiple input, which is larger than the sum of their individual responses (i.e. supralinear), will enhance sensitivity (*Rubin et al., 2015*) and therefore this mechanism may constitute an important physiological purpose.

**Table 1.** Two regimes of neuronal spiking and their definition, properties and causes.

|  | Fluctuation–driven | Mean–driven | Key references |
|---|---|---|---|
| Definition | $R_m I_{total} < V_{thres}$ | $R_m I_{total} > V_{thres}$ | (*Gerstner et al., 2014*; *Brunel, 2000*) |
| Properties | Lower firing rates | Higher firing rates |  |
|  | Irregular spiking | Regular spiking | (*Amit and Brunel, 1997*; *Shadlen and Newsome, 1998*; *van Vreeswijk and Sompolinsky, 1998*) |
|  | Lognormal/Skewed distribution | Symmetric distribution | (*Buzsáki and Mizuseki, 2014*) |
|  |  |  | (*Roxin et al., 2011*; *Mizuseki and Buzsáki, 2013*) |
| Cause | Balanced E/I | Intrinsic currents, unbalanced E/I | (*Bell et al., 1995*; *Shadlen and Newsome, 1994*; *Softky and Koch, 1993*) |
|  | Synchronized excitation |  | (*Stevens and Zador, 1998*) |

This is in contrast to the mean–driven regime where the summation is linear or even sublinear, which will transform a normally distributed input to a normally (as opposed to lognormally) distributed firing rate (*Figure 1B*). Such linear (or sublinear) transformation will reduce rather than enhance sensitivity and therefore the mean–driven regime will curb network activity (*Ahmadian et al., 2013*). These two transformations work together into an S-shaped IO-curve, where weak input are amplified yet the network is kept stable for strong activity (*Figure 1C*). Sample neurons in the two regimes are shown (*Figure 1D–E*). If this mechanism is true, then the shape of the firing rate distribution will reveal the spiking regime of a given neuron. The degree to which neurons operate in one versus the other regime may hold the key to understanding stability, dynamic range and other important properties of network operations. Yet this still remains to be investigated, especially in spinal networks.

Here, we investigate the regimes of operation of spinal neurons during different rhythmic motor behaviors, which are generated in the lumbar spinal circuits of turtles. We test the theoretical scheme put forward by *Roxin et al. (2011)*, by assessing the synaptic input, the spike response function in subthreshold domain, and determine the shape of the firing rate distribution. The mechanical stability of the turtle preparation allows electrophysiological recordings of unprecedented quality, such that we can combine intracellular recording with multi–electrode arrays, and thus determine the fraction of the population in the two regimes at all times. The high resistance to anoxia of turtles allows using adult animals with fully developed spinal circuitry, which have healthy network activity and which can perform multiple complex motor behaviors (*Stein, 2005*). Thus, we can investigate the population activity during, not just one behavior, but multiple motor behaviors. Custom designed high–density silicon electrodes recorded the population activity from hundreds of cells in the dorsoventral and rostrocaudal axes along with the intracellular $V_m$ of single neurons and multiple relevant motor nerves (*Figure 2*). This is a unique experimental investigation, because it explores the link between neuronal ensemble data, which in itself is rare in spinal motor research, and the forefront of theoretical neuroscience.

## Results

The parallel spiking activity of 200–300 single units were recorded in the medial to ventral horns of lumbar spinal segments involved in motor rhythm generation (*Figure 2A*). The location of the electrode arrays in the ventral area of the lumbar enlargement was verified by histology (*Figure 2B–C* and *Figure 2—figure supplement 1*). The array recordings were performed simultaneously with recording of the intracellular activity of a single neuron in parallel with electroneurograms (ENGs) from relevant motor nerves (*Figure 2D*). Site–specific rhythmic hindlimb scratching was induced by tactile touch of the carapace (*Berkowitz et al., 2010*; *Stein, 2005*) and could be reproduced reliably over multiple trials (*Petersen et al., 2014*; *Vestergaard and Berg, 2015*). The extracellular multi-electrode arrays, which were used, were custom–designed for the spinal cord (Berg64-probe, Neuronexus inc.) to enable efficient polytrode spike sorting (*Figure 2E* and *Figure 2—figure supplement 2*). The distribution of spike count firing rates across the population was skewned (*Figure 2F*), but resembled a normal distribution on logarithmic x-axis (inset), i.e. a lognormal distribution. This

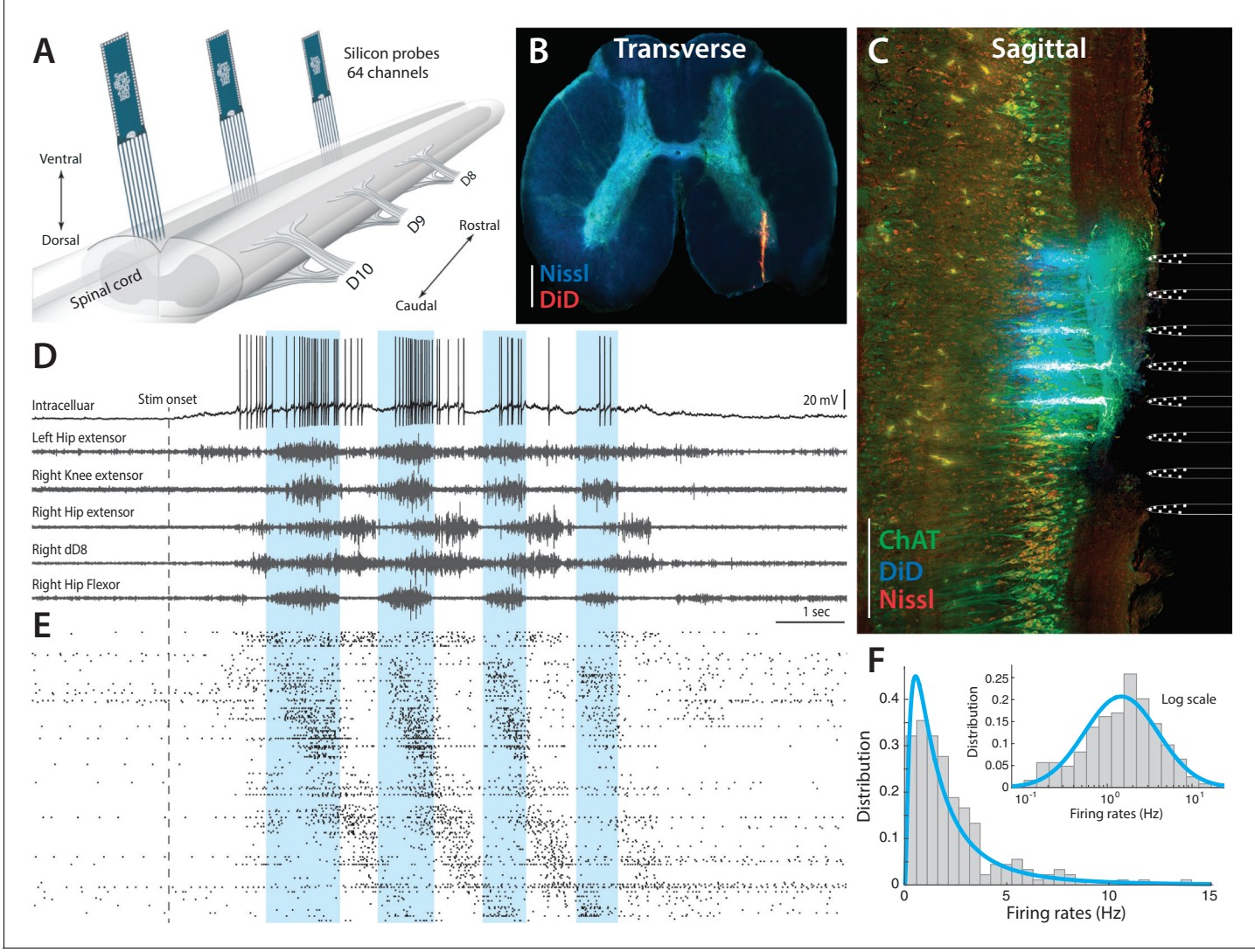

**Figure 2.** Parallel neuronal activity in the lumbar enlargement during rhythmic motor activity. (A) Illustration of experiment with three silicon probes inserted into the lumbar spinal cord of a turtle. Histological verification: transverse (B) and sagittal (C) slices, 200 µm thick, showing the location of the silicon probes in the spinal cord (red traces and location illustrated on right, electrodes stained with DiD). ChAT staining in green and Nissl stain in blue. Scale bars: 500 µm (D) $V_m$ of a single neuron (top) concurrently recorded with five motor nerves (traces below) during scratching behavior induced by a somatic touch (onset indicated, 10 s duration). (E) Rastergram showing the parallel-recorded single units (∼200 neurons) sorted according to hip flexor phase. (F) Firing rate distribution is positively skewed and normally distributed on a log–scale, i.e. lognormal (inset). $V_m$ resting level in (D) is −60 mV. For details, see *Figure 2—figure supplement 1* and *2*.

The following figure supplements are available for figure 2:

**Figure supplement 1.** Experimental setup.

**Figure supplement 2.** Sorted sample units, quality measures, and probe layout.

lognormal distribution indicates a wide degree of participation in the motor activity across the population. In the following, we will investigate the participation of neurons within the mean– and fluctuation–driven regimes and how this is linked to the lognormal firing rate distribution, both across the population and within individual cells. We start by addressing the mechanism behind the lognormal firing rate distribution in intracellular recorded data, before addressing the concurrent population activity.

## Mechanisms behind lognormal distribution and the fluctuation–regime

Two mechanisms have previously been proposed to explain the skewned lognormal firing rate distribution, which is also observed in other parts of the nervous system (*Buzsáki and Mizuseki, 2014*). Lognormal distributions could either arise from a nonlinear transformation of normally distributed inputs (*Roxin et al., 2011*) (*Figure 1A*) or from a linear transformation of a lognormally distributed synaptic input (*Wohrer et al., 2013*). The latter mechanism was considered in connection with the sparse spiking activity in auditory cortex (*Koulakov et al., 2009*; *Hromádka et al., 2008*) and since synaptic weights within neocortex have a heavy tail lognormal distribution rather than a Gaussian distribution (*Ikegaya et al., 2013*; *Song et al., 2005*). Models also show that the $V_m$ distribution can be either skewed or Gaussian depending on the synaptic input intensity (*Ostojic, 2011*). Therefore, to distinguish between the proposed mechanisms, it is important to first assess whether the synaptic current is normally versus lognormally distributed. Secondly, to test whether the transformation of the synaptic input to spiking output is linear versus supralinear. We started by addressing the first requirement by investigating the synaptic input in intracellular recordings. The most relevant part of the data was found during the peak of a locomotor cycle where the $V_m$ was in vicinity of $V_{thres}$ and was dominated by synaptic potentials (*Figures 1D* and *3A*). The motor activity was clearly non–stationary, which means that the spike activity was likely to move between the fluctuation– and mean–regime. Nevertheless, the rhythmic activity possessed a separation of timescales in the sense that the activity between cycles ($\sim$1 s) contained much larger excursions in $V_m$ than within cycles ($\sim$2-400 ms). Here, the mean $V_m$ did not change much and for practical purposes it could be considered constant within the cycle. In the following analysis of the intracellular data we regarded the dynamics in $V_m$ as stationary within a cycle – well aware that the comparison to theoretical models, which are based on assumption of stationarity, should be taken with a grain of salt. We intended to investigate the symmetry of the distribution of synaptic current using this assumption. The synaptic current within a cycle is difficult to assess, but rather than the mean current, we were primarily interested in the fluctuations in current, which we could approximate from $V_m$ via Ohm's law under the following conditions. Within a cycle, the mean $V_m$ was just below threshold and did not change its value much. Therefore the voltage–activated conductances were approximately constant such that there was an Ohmic relationship between synaptic current and $V_m$. This is likely justified for neurons in fluctuation–driven regime, since the conductance is often high and dominated by balanced E/I synaptic input (*Destexhe et al., 2003*; *Kumar et al., 2008*). The high conductance suppresses the coupling between $V_m$ and intrinsic conductance in a divisive manner (*Kolind et al., 2012*; *Tiesinga et al., 2000*). Thus, in the fluctuation–driven regime the non–Ohmic contributions were likely smaller and the $IV_m$-relationship more linear than in the mean–driven regime.

## Normally distributed synaptic input

We intended to test the hypothesis of normally distributed input, but since the approximation of using the variability in $V_m$ as a proxy for the variability in synaptic current is most valid for the neurons in fluctuation–driven regime, we needed a way to distinguish neurons that were primarily in the fluctuation–driven regime. We therefore propose a novel metric, the return map ratio , which quantifies the degree of fluctuations leading up to a spike (*Figure 3—figure supplement 1*). The return map ratio (RMR) quantifies how direct the subthreshold $V_m$–trajectory is between spikes and this forms the basis for selecting neurons in our analysis. An RMR close to 0.5 has fluctuation–driven spiking whereas a value close to 1 has mean–driven spiking (*Figure 3—figure supplement 1A,B*). Therefore, we defined a neuron as fluctuation-driven if its RMR <0.7; in our sample of intracellular recordings we found 50/68 neurons in this regime. A sample neuron, which was found in the fluctuation–driven regime based on this metric illustrates how we obtained the distribution of sub–threshold $V_m$ (*Figure 3A*). The distribution was estimated both by selecting the $V_m$ in between spikes (temporal distribution) and by collecting instances of $V_m$ prior to spike peak in a spike triggered overlay ('sigma' in *Figure 3B*). These two estimates are in agreement with one another for the sample cell (*Figure 3C*). This agreement is also found across the population as quantified by the mean and SD (*Figure 3D*). The skewness for the distributions across the population is small and scattered around zero as expected for normal (symmetric) distributions (*Figure 3E*). From these data we conclude that the subthreshold $V_m$–distributions are not skewed, but rather symmetrical and *Gaussian–like* (cf. inset distributions, *Figure 3E*). Nevertheless, the minimal requirement for confirming the

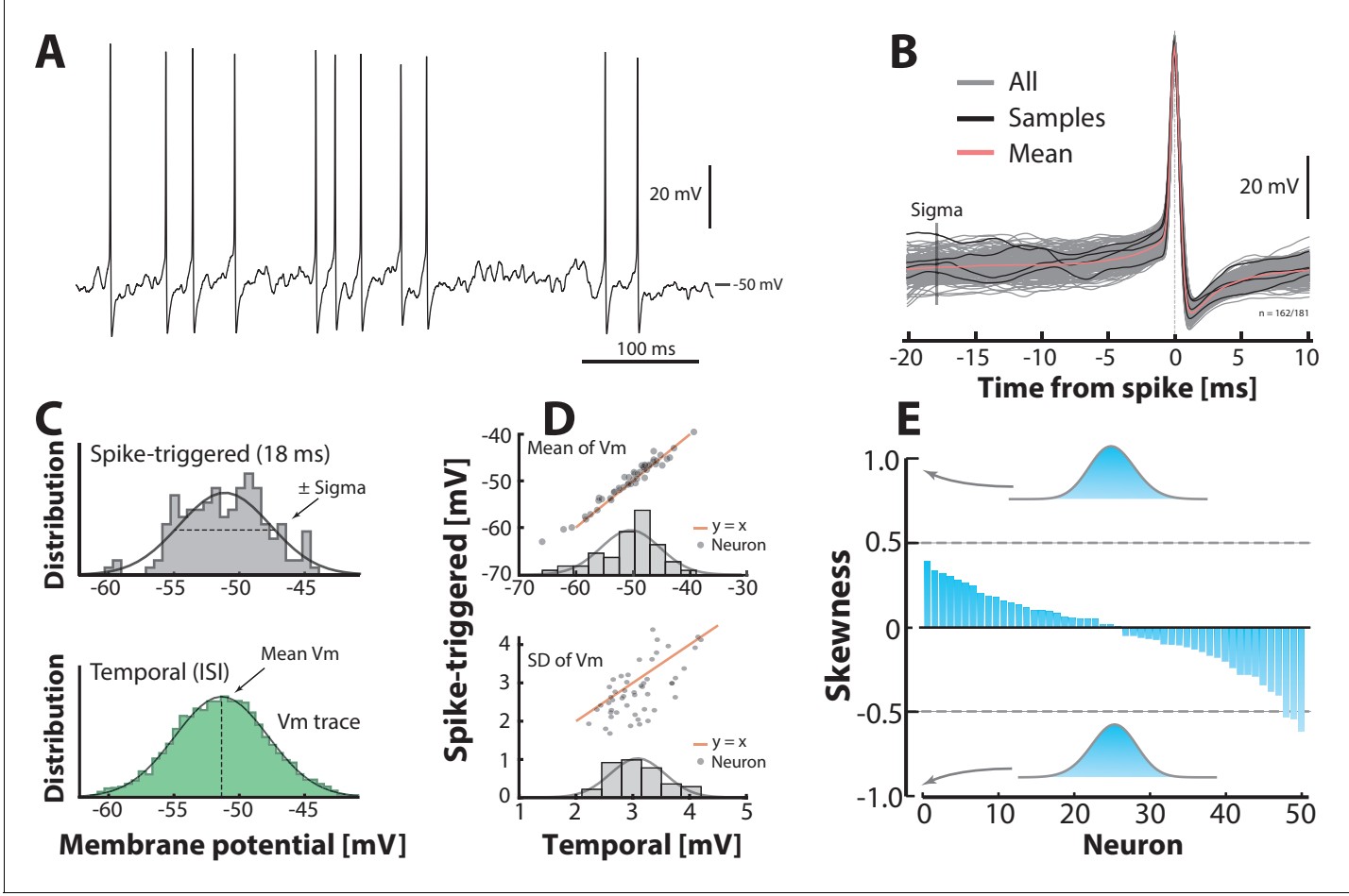

**Figure 3.** Subthreshold $V_m$–distributions are symmetric. (**A**) Sample cell spiking in the fluctuation–driven regime, and (**B**) its spike–triggered overlay to determine the $V_m$–distribution of trajectories 18 ms prior to spike–onset ('sigma'). (**C**) The $V_m$–distribution is estimated in two ways: via samples of $V_m$–instances prior to the spike peak (top, vertical line 'sigma' in **B**) and over time via the interspike intervals (bottom). (**D**) Mean temporal– vs. spike–triggered–estimates (top) are closely related (orange unity–line) and have a near normal distribution of means (inset). For details, see *Figure 3—figure supplement 1* and *2*. Similarly, the variability of the two estimates (SD) are closely related (bottom). (**E**) Sorted skewness for all neurons in fluctuation–driven regime indicate symmetric $V_m$–distributions (temporal). Inset distributions with skewness of $\pm 1$ illustrate no discernible asymmetry. The extreme skewness observed in the data set is around $\pm 0.5$ (broken lines).

The following figure supplements are available for figure 3:

**Figure supplement 1.** Quantifying the degree of fluctuations and selecting neurons in fluctuation–driven regime using the return map ratio metric.

**Figure supplement 2.** Population–distribution of mean $V_m$ is Gaussian.

two–regime hypothesis for the single neuron is that the synaptic current (not the synaptic potentials) is Gaussian (*Figure 1*). As we argued earlier, if there is an Ohmic relationship between current and potential, which is likely during high–conductance states, then this requirement would be granted. More importantly, now that we do find a Gaussian $V_m$–distribution, it is difficult to contemplate a non-linear $IV_m$-relationship, which would result in such a symmetric distribution. The synaptic input current would have to have a finely matched inverse distribution to cancel out this non–linearity in order to achieve a symmetric $V_m$–distribution. A more parsimonious explanation therefore is that, since the synaptic potentials are normally distributed, they are a result of a linear transformation of synaptic currents, which are also normally distributed.

So far, we have only looked at $V_m$–distributions of single neurons, which operate primarily in the subthreshold domain, and found that the synaptic input is most likely normally distributed. We do

not know whether the synaptic input is also normally distributed in the mean–driven regime, but since the synaptic input is normally–distributed in the subthreshold region, it is likely also normally–distributed in the suprathreshold region. Otherwise, the input statistics from the presynaptic neurons would have to depend on the threshold of the post–synaptic neuron, which is unlikely.

## Mean $V_m$ across the population is normally distributed

Above, we established that the synaptic input to a given neuron is likely normally distributed, and if this input is transformed in a supralinear fashion, the output firing rate distribution will be skewed. Nevertheless, the foundation of the skewness in population rate distribution (*Figure 2F*) is not necessarily directly linked to the skewness of the instantaneous rate distribution of single neurons. In principle, it is possible to have a population with a normal distribution of mean firing rates, where the cells themselves have lognormally distributed firing rates and vice versa. Therefore, we needed to address the distribution of mean $V_m$ across the population and test whether this was skewed or normal. Further, since the sub–threshold IO-curve is linked to threshold, it is important to establish the distance of mean $V_m$ from threshold with respect to the size of synaptic fluctuations, i.e. standard deviation of $V_m$ ($\sigma$). This distribution, i.e. $(V_m - V_{thres})/\sigma$, turns out to also be normally distributed with a mean around $3\,\sigma$ from threshold (*Figure 3—figure supplement 2*, plotted for all $n = 68$ neurons). The value used for $V_{thres}$ here is the mean of the estimated thresholds for all spikes (see below). If we assume, when normalizing $V_m$ this way, the IO-curve has approximately the same nonlinearity across all neurons, the population distribution of firing rates will also be skewed due to the nonlinear transformation of the normally–distributed input (*Figure 3—figure supplement 2F*) to a lognormally–distributed output. These results are in qualitative accordance with the scheme proposed previously (*Roxin et al., 2011*). As another piece of the puzzle, we need to establish the shape of the neuronal response function, which rarely has been done in the subthreshold domain.

## Neuronal response–function in subthreshold domain is nonlinear.

The link between a normally distributed input and a lognormally distributed output is a supralinear transformation. To test whether this is a hallmark of the fluctuation–driven regime, we needed to estimate the input–output (IO)–function for the subthreshold domain. The IO–function of neurons is a fundamental property of the nervous system, and therefore it is well-characterized both theoretically (*Gerstner et al., 2014*) and experimentally (*Silver, 2010*). Nevertheless, it has rarely been established for fluctuation–driven spiking. Here, we estimated the IO-function for subthreshold spiking via the probability of eliciting a spike as a function of $V_m$ in the following way. First, we collected instances of $V_m$ shortly before the spike–onset, where $V_m$ is depolarized yet still not part of the deterministic spike trajectory. The probability that a given value of $V_m$ will cause a spike was estimated as the histogram of $V_m$–instances (gray histogram, *Figure 4A*) divided by the total time spent at all values of $V_m$ (green histogram). This gives the empirical relationship between $V_m$ and the firing rate (*Jahn et al., 2011*; *Vestergaard and Berg, 2015*). The IO–function had a strong non–linear shape (*Figure 4B*). To capture the curvature we fitted both a power–law and an exponential for all $n = 68$ neurons and the curvature had a weak negative correlation with the SD of the $V_m$–fluctuation (*Figure 4C–D*) as demonstrated previously (*Vestergaard and Berg, 2015*). Similar expansive nonlinearity has previously been characterized in sensory–driven neurons (*Anderson et al., 2000*; *Hansel and van Vreeswijk, 2002*; *Miller and Troyer, 2002*). It will transform the normally–distributed synaptic potentials into a lognormally–distributed spiking output in the fluctuation-driven regime (*Figure 1A*). For mean–driven spiking the IO-function is not supralinear, but rather linear (or even sublinear), and the normally–distributed synaptic input will therefore be transformed to a normally distributed spiking output (*Figure 1B*). In conclusion, neurons that have fluctuation–driven spiking also have a non–linear IO-transformation of synaptic potentials to spiking output.

## Lognormal firing rate distribution in single neurons

The normally distributed input combined with the nonlinear IO–transformation should result in a skewed lognormal firing rate in the single neuron. To confirm this, we measured the distribution of the instantaneous firing rate, i.e. the inverse of ISIs. The quiet period in between burst cycles were not included in the analysis (*Figure 1D–E*), since in these periods $V_m$ was far from $V_{thres}$ and therefore in an irrelevant part of the IO–function. The firing rate distribution of many cells was positively

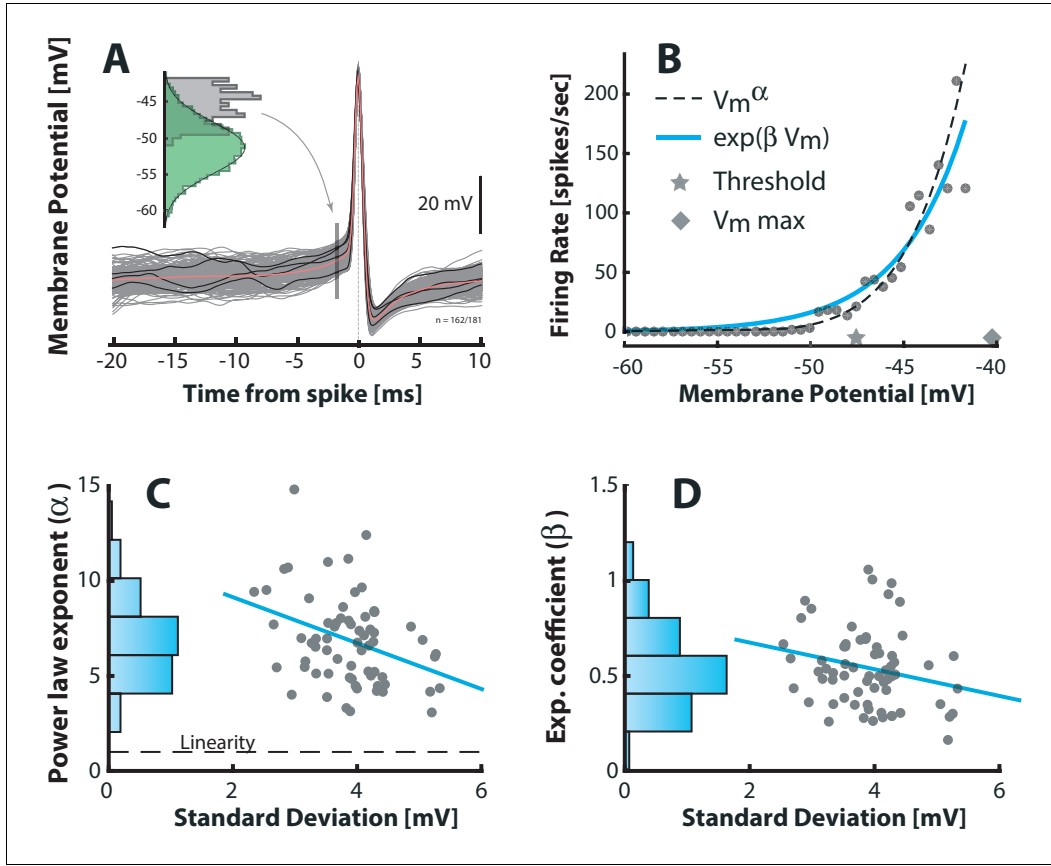

**Figure 4.** Fluctuation–driven spike–response curve is supralinear. (A) The empirical probability of evoking a spike in a small window as a function of $V_m$ is determined using spike–triggered overlays. The probability distribution is estimated as the $V_m$–distribution of trajectories prior to spike–onset (gray histogram, 1.7 ms prior to peak) normalized with the total (temporal) $V_m$–distribution (green histogram). Dividing this probability by the sampling interval gives the firing rate (see Materials and methods). (B) The firing rate versus $V_m$ for a sample neuron is strongly nonlinear. A power–law (broken line) and an exponential (blue line) are fitted to capture the nonlinearity. Note that the mean threshold (★) is below the largest subthreshold fluctuation (◇), likely due to a depolarization of threshold associated with a higher firing rate (see also **Figure 6—figure supplement 1**). (C) Power–law exponent ($\alpha$) for different neurons are weakly anti–correlated with the fluctuations (SD) in their $V_m$ ('sigma', **Figure 3B**, $R = -0.34$, p<0.01). Linearity is indicated by horizontal broken line. (D) Exponential coefficient ($\beta$) for different neurons are also anti–correlated with the fluctuations in $V_m$ albeit not significantly ($R = -0.22$, p>0.05).

skewed and resembled a normal distribution with near zero skewness on a log-scale (sample cell shown in **Figure 5A**). This is expected for poisson–like spiking in the fluctuation–driven regime (**Ostojic, 2011**). Nevertheless, distributions for all the intracellularly recorded neurons ($n = 68$) were skewed to a varying degree from strong positive to zero skewness on a linear axis and similarly shifted downwards on log axis (cf. gray and green histograms, **Figure 5B**). This suggests that neurons were found in a spectrum between fluctuation– and mean–driven spiking. More negative log–skewness were associated with higher mean rates (**Figure 5C**). This is probably due to a larger presence in the mean–regime at higher firing rates, where the distribution skewness is expected to be negative on a log–scale, i.e. Gaussian on a linear scale. Note that the spectrum of skewness was substantially larger than it was for the $V_m$ distributions above (**Figure 3E**). Skewed Gaussian distributions are shown to illustrate the range of skewness in the data (**Figure 5D**). In conclusion, these results suggest that the skewness in firing rates is an indicator of the degree of participation in the fluctuation–driven regime.

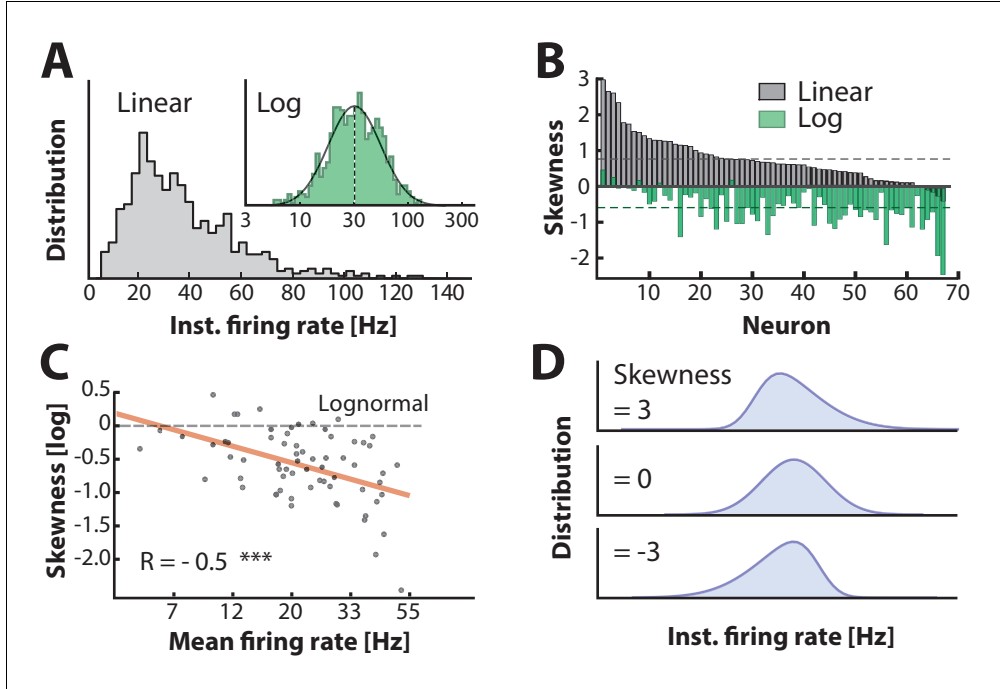

**Figure 5.** Firing rate distributions are skewed to a variable degree depending on mean firing rate. (**A**) Distribution of instantaneous firing rates for a sample neuron is positively skewed on a linear axis and lognormal–like (green histogram, inset). Mean indicated by broken vertical line. (**B**) Sorted distribution skewness on linear (gray) and logarithmic axes (green) for each neuron in the population. (**C**) The log–skewness across neurons is negatively correlated ($R = -0.5$, p<0.001) with mean firing rate, which indicates that higher firing rates are found in the mean–driven regime and less lognormally–distributed, i.e. departing from broken line. (**D**) Illustration of firing rate distributions that have positive skewness (top), zero skewness (Gaussian, middle) and negative skewness (bottom) representing the range observed in the data (**B**).

## Time spent in regimes: intracellular data

A neuron is not just spiking in either the fluctuation– or the mean–driven regime, rather, it likely spends time in both regimes during motor activity. To estimate the amount of time a given neuron spends in either of the two regimes we calculated the fraction of time that the smoothed $V_m$ was above versus below threshold. We first look at two heuristic neurons, one in the fluctuation–driven regime and one in the mean–driven regime. The fluctuation–driven neuron spent most of the time below threshold (**Figure 6A**) and had more irregular spiking as quantified by a local measure of irregularity, the $CV_2$ (green line). $CV_2$ is the difference of two adjacent ISIs divided by their mean (**Holt et al., 1996**; **Bruno et al., 2015**). In contrast, the mean–driven neuron spent most time above threshold and had more regular spiking, i.e. $CV_2$ closer to zero (**Figure 6B**). Since the threshold was firing rate–dependent due to the inactivation of the $Na^+$–conductance (**Figure 6—figure supplement 1**) we used the most hyperpolarized value of threshold (broken line). The distribution of $CV_2$ for all trials had higher mean for the fluctuation–driven cell than the mean–driven (cf. arrows, **Figure 6C**). Also, the cumulative time spent below threshold was higher for the fluctuation–driven cell (96%) than the mean–driven cell (35%, **Figure 6D**). This fraction of time spent below threshold was quantified for every neuron ($n = 68$) and the population distribution had a strong mode at 1 (top, **Figure 6E**) suggesting many neurons spent much time in the fluctuation–driven regime. To compress the diversity within the population into a simpler representation, we used the reverse cumulative distribution of neurons versus time spent below threshold (bottom, **Figure 6E**). This indicates how many neurons (y-axis) spent at least a given fraction of time (x-axis) below threshold. The intercept with the 50%–line (broken line) indicates what fraction of time half the population at least spent below threshold. This fraction is remarkably high (84%) suggesting a prominent presence within the fluctuation–driven regime.

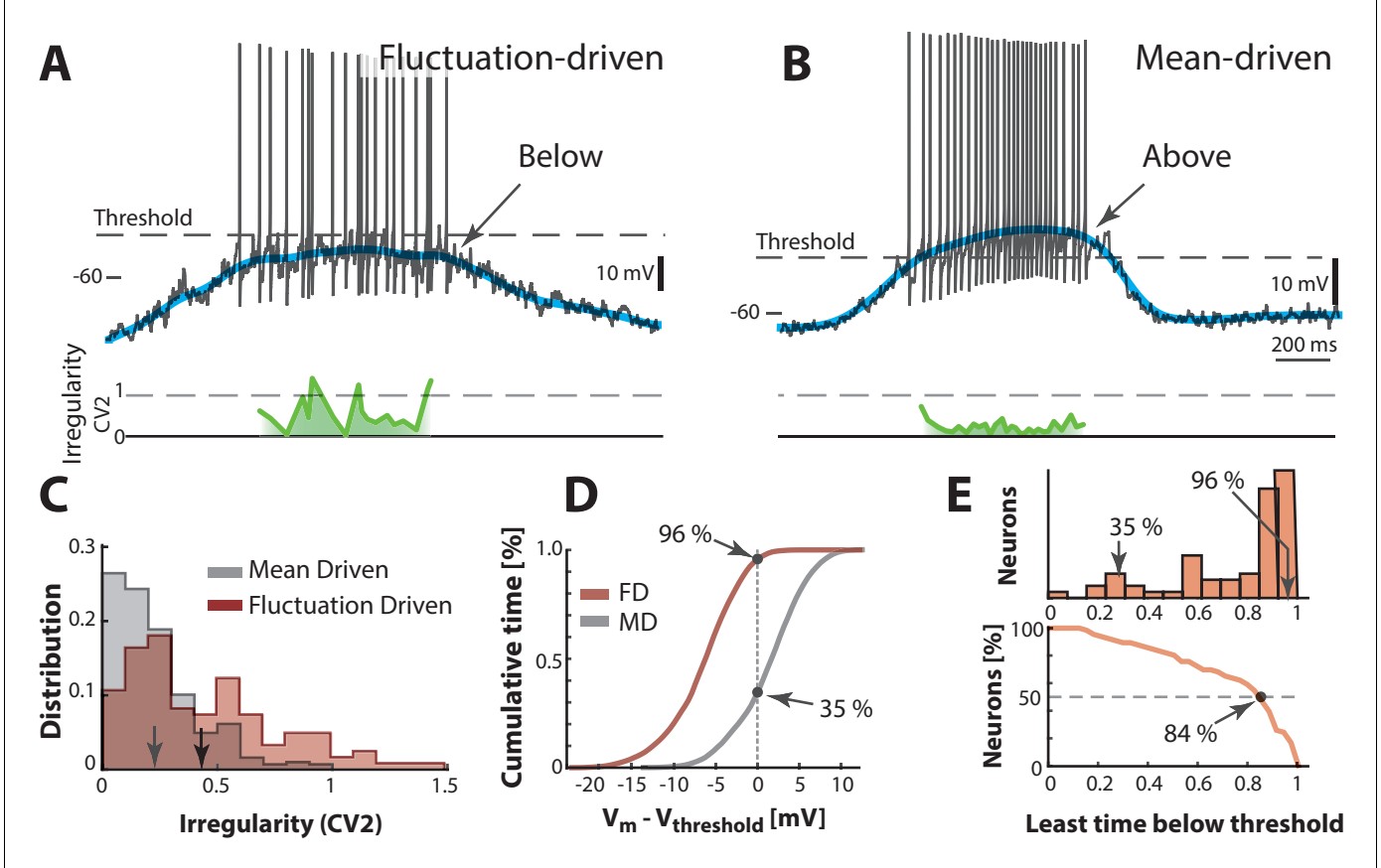

**Figure 6.** Two contrasting sample neurons found in the two regimes. (A) Sample neuron in fluctuation–driven regime, where the mean $V_m$ (blue line) is below lowest threshold (broken line), the spikes are irregular ($CV_2 \approx 0.5$–1, green line) and driven by fluctuations (arrow). (B) Second sample cell found in mean–driven regime, where the mean $V_m$ is above threshold during the cycle (arrow). The spiking is more regular, i.e. low $CV_2$ (green line). (C) Mean–driven neuron (gray) has lower $CV_2$ than the fluctuation–driven neuron (brown). Means indicated (arrows). (D) Cumulative time of $V_m$ shows the fluctuation–driven neuron (FD) spends more time below threshold (96%) than the mean–driven (MD, 35%). (E) Top: Time below threshold for population of neurons (cells from A–D indicated). Bottom: Least time spent below threshold versus a given fraction of neurons (reverse cumulative distribution function). Half of the neurons (broken line) spend at least 84% of the time in fluctuation–driven regime, i.e. have $V_m$ below threshold.

The following figure supplement is available for figure 6:

**Figure supplement 1.** Threshold depolarizes with increase in firing rate.

## Transition between regimes by current injection

Mean- and fluctuation-driven spiking can be distinguished by important traits such as degree of irregularity and log-skewness of the firing–rate distribution. To verify these traits, we used another sample neuron as a heuristic illustration. We injected different levels of either positive or negative bias currents in different trials while keeping all else constant. A negative constant current injection ($-1.0$ nA) caused a decrease in firing rate and a slight increase in irregularity (green line) compared with zero injected current (*Figure 7A–B*). Similarly, a positive current injection (1.7 nA) caused more spikes and a decrease in irregularity (*Figure 7C*) consistent with a movement between regimes (inset in *Figure 7A*). The decrease in irregularity with increasing input was further quantified as a negative correlation between mean $CV_2$ and injected current ($R = -0.84$, p≪0.001) over multiple trials (n=18, *Figure 7D*). This is qualitatively in agreement with previous reports (*Prut and Perlmutter, 2003*; *Powers and Binder, 2000*; *Wohrer et al., 2013*). The instantaneous firing rate in the control condition (0 nA) was lognormal as expected for the fluctuation–driven regime (top, *Figure 7E*). When adding input current the distribution was shifted to the right and enriched with a negative skewness as

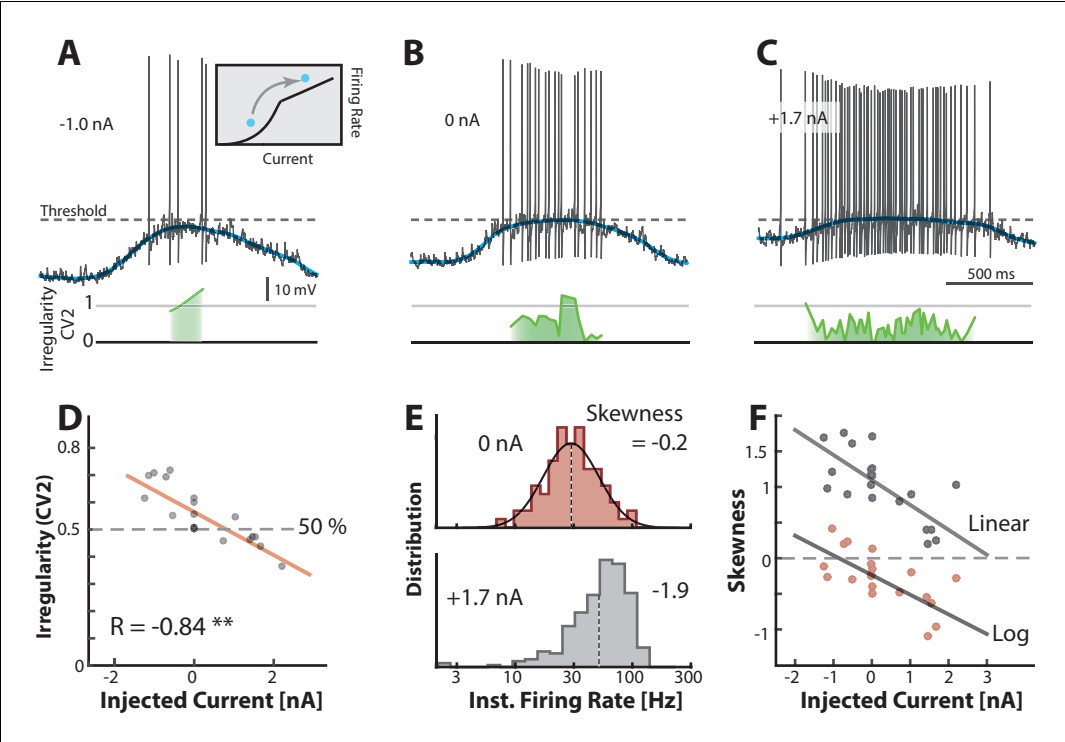

**Figure 7.** Transition between regimes can be induced by injected current. (**A**) Hyperpolarizing $V_m$ of a sample neuron during the motor cycle with negative injected current (−1.0 nA). Negative current hyperpolarizes mean $V_m$ (blue) and increases irregularity ($CV_2 \approx 1$, green line) compared with control condition (**B**). (**C**) Positive current injection (1.7 nA) has the opposite effect: Depolarization, more regular spiking and higher firing rate. (**D**) Mean of $CV_2$ over a trial vs. the constant injected current for that trial has negative correlation. (**E**) Firing rate is lognormally distributed in control (top), but negatively skewed (skewness = −1.9 when added current increases mean–driven spiking (bottom). (**F**) Skewness of firing rate distribution is negatively correlated with injected current. Linear skewness shown in top gray points ($R = -0.73$, p<0.001) and log-skewness shown in bottom red points ($R = -0.70$, p<0.001). Same neuron throughout.

expected for mean-driven spiking (bottom, *Figure 7E*). This relation between input and shape of rate distribution was further confirmed by a negative correlation between multiple current injections and skewness both on linear scale (gray dots) and log–scale (red dots, *Figure 7F*). Hence, skewness and irregularity are indicators of the spiking regime.

## Blocking inhibition causes change in regime

An alternative to injecting electrode current is to manipulate the balance of excitation and inhibition (E/I) by pharmacological means. This is important for understanding the cause of irregularity and the fluctuation–driven regime. Hence, we manipulated the synaptic input in a reduced preparation with micro–superfusion of strychnine, a glycinergic blocker, over the transverse cut surface of the spinal cord (described in [*Berg et al., 2007*; *Vestergaard and Berg, 2015*]). This affected only neurons at the surface (<300 μm) without affecting the rest of the network, which was verified by careful monitoring of flow and the network activity via the nerve recordings. Comparing the spiking during control condition (*Figure 8A*) with that during blockade of inhibition (*Figure 8B*), we noticed a strong increase in spiking. This is consistent with a depolarization due to disinhibition, thus 'unbalancing' the excitatory and inhibitory input. Reducing inhibition tipped the balance of E/I toward larger inward synaptic current, which resulted in a more depolarized $V_m$ (blue line) well above threshold (arrows, *Figure A–B*). It also resulted in higher firing rates and lower irregularity on the peak (cf. green lines). Generally, the irregularity ($CV_2$) was higher in the control case than in the unbalanced case (*Figure 8C*) similar to the results observed with current injection (*Figure 7A–D*). The irregularity

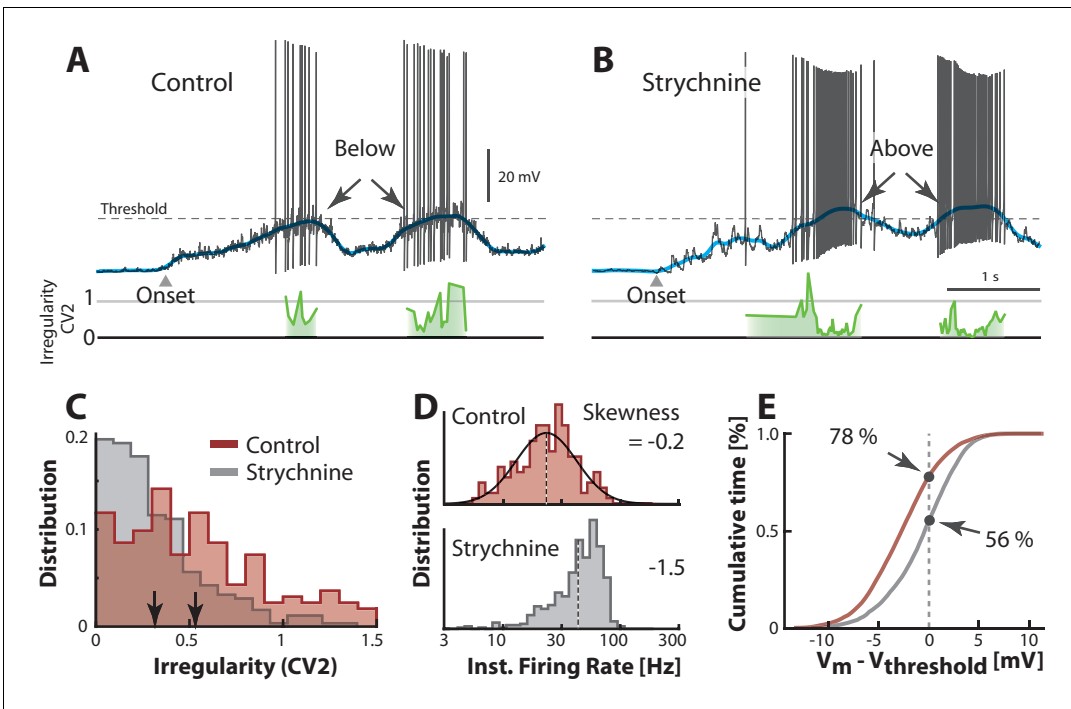

**Figure 8.** Transition between regimes induced by unbalancing E/I. (A) Sample cell in control condition and after reduction of inhibition with local strychnine (B). Onset of motor program indicated (△). Blocking inhibition results in a larger net inward current, which drives the mean $V_m$ (blue lines) across threshold to more mean–driven regime. As a result the spiking is less irregular on the peaks as measured with $CV_2$ (cf. green lines). (C) Irregularity ($CV_2$) was smaller after application of strychnine (arrows indicate mean, histogram truncated at 1.5). (D) Firing rate distribution is symmetric on log–scale (top, skewness = −0.2) and negatively skewed when inhibition is blocked (bottom, skewness=−1.5). (E) Strychnine induces a more depolarized Vm and a lower cumulative time spend below the threshold (compare 78% with 56%). Same neuron throughout.

The following figure supplement is available for figure 8:

**Figure supplement 1.** Unbalancing E/I induces an anti–correlation between irregularity and depolarization.

was also negatively correlated with depolarization of the mean $V_m$ when unbalancing the E/I although it was uncorrelated in the control condition, where the spiking occurred in the fluctuation–driven regime (*Figure 8—figure supplement 1*). The instantaneous firing rate was skewed and log-normal in the control case (top, *Figure 8D*), similar to the above sample cell (top, *Figure 7E*). This distribution became negatively skewed when adding inward current (bottom, *Figure 7E*). Similar effect was seen when 'unbalancing' the synaptic input, which also result in larger inward current. The firing rate increased (cf. broken lines, *Figure 8D*) and the distribution became negatively skewed (cf. −0.2 and −1.5) as expected in the mean–driven regime (bottom). To quantify the increase in time spent in the mean–driven regime, we performed an analysis similar to the analysis in the above section (*Figure 6D*). The cumulative time spent below threshold was larger in the control condition (78%) compared with the unbalanced case (56%, *Figure 8E*). These observations are largely consistent with the consensus view that irregular fluctuation–driven spiking is due to a balance between excitation and inhibition (*Table 1*).

## $CV_2$ as an indicator of spiking regime

In the above intracellular analyses we reported the spiking irregularity in terms of $CV_2$ along with the mean $V_m$, current injection and pharmacological manipulation of the balance of excitation and inhibition. The $CV_2$ measure is convenient to use as an indicator of the mean– versus the fluctuation–driven

regimes observed in the extracellular spiking data, since it only requires spike times. Therefore it is important to validate $CV_2$ as an indicator of spiking regime. In the above sample cell analyses we note first, that when $V_m$ spent a larger fraction of time above threshold, i.e. in mean–driven regime, the $CV_2$ was lower (*Figure 6*). Second, when depolarizing a neuron artificially either with constant positive current (*Figure 7D*), or by blocking inhibition (*Figure 8C*), such that more spikes were in mean–driven regime, the $CV_2$ was decreased.

To further substantiate $CV_2$ as an indicator of spiking regimes we looked again at the return map ratio, which is an independent metric of fluctuations during inter-spike intervals. If $CV_2$ is an indicator of the spiking regime, it should be anti-correlated with the return map ratio. This was confirmed by plotting the mean $CV_2$ for all cells ($n = 68$) against the mean return map ratio, which indeed demonstrated a significant anti–correlation ($R = -0.34$, p=0.005) (*Figure 3—figure supplement 1E*).

A second independent indicator of fluctuation regime is the cumulative time below threshold of $V_m$ (*Figure 6D*), which should be correlated with the mean $CV_2$. We tested this using the most hyperpolarized value of theshold, since it was the most conservative, but there was no significant correlation between the cumulative time below threshold and the mean $CV_2$. Perhaps the lack of linear relationship is due to a bias from the reset voltage and after-hyperpolarization, which is different from cell to cell and therefore randomly may introduce a large fraction of time spent below threshold. Also, intense synaptic activity is known to quench the after–hyperpolarization (*Berg et al., 2008*) and therefore this bias may be particularly strong when the synaptic input is not balanced as in the mean–driven regime.

A third indicator of spiking regime is the skewness of the instantaneous firing rate distribution (*Figure 7E* and *8D*). We estimated the skewness of the individual firing rate distributions for all neurons ($n = 68$) and plotted it against the mean $CV_2$ (data not shown). There was a significant positive correlation between the two, regardless of whether the firing rate distribution was plotted on log or linear scale ($R_{log} = 0.43$, p=0.0003, and $R_{lin} = 0.41$, p=0.0006), which suggest $CV_2$ as a valid measure for spiking regimes.

A last indicator is the local mean membrane potential depolarization, which should be anti-correlated with the instantaneous $CV_2$, if the $V_m$ is above threshold (*Figure 8*, *Figure 8—figure supplement 1D*). Here, there was a lack of correlation between $CV_2$ and $V_m$ before blocking inhibition, in the fluctuation–driven regime. However, after removal of inhibition, $V_m$ was in supra–threshold domain, which introduced an anti-correlation between $CV_2$ and $V_m$. Hence, if the neuron is in the mean-driven regime the $CV_2$ is an indicator for the depolarization above threshold. To further verify this we performed a similar test of the relationship between instantaneous $CV_2$ and local depolarization for all neurons (without pharmacology). We found that all the cells with significant relationships (p<0.05, $n = 16/68$) had anti-correlation between $V_m$ and $CV_2$ (data not shown). In conclusion, the $CV_2$ measure is correlated with other measures and indicators of spiking regimes (except the cumulative time below threshold) and therefore $CV_2$ is a useful indicator in itself.

## Noisy threshold has no effect

The irregularity in spiking could be caused by a noisy threshold rather than fluctuations in synaptic potentials. Nevertheless, a noisy threshold can only explain a small part (if any) of the spiking irregularity. First of all, if the irregularity, that we observed in spike times, was due to a noisy threshold mechanism, we should see the same irregularity regardless of the depolarization, i.e. regardless of whether the neuron was in the sub–threshold or supra–threshold domain. Yet, the spiking irregularity was strongly dependent on depolarization (*Figures 6–8*). There was an adaptation in threshold (*Figure 6—figure supplement 1*). This was not random, but rather due to a gradual inactivation of Na$^+$–channels throughout the burst (*Henze and Buzsáki, 2001*). The threshold of a given spike strongly depended on the threshold of the previous spike (panel F) as well as the mean firing rate (panel G). The same mechanism is behind spike–frequency adaptation, which is a well–described phenomenon (*Grigonis et al., 2016*). The adaptation in threshold is likely to make the IO-function more sublinear in the mean–driven regime, which will generally curb network activity.

In order to verify the extent of the threshold variance beyond the contribution from inactivation of Na$^+$–channels, we looked at the threshold of only the first spike of each cycle, such that the neuron had ample time for recovery. The variance of the first–spike threshold ($n = 51$) in a sample neuron was $\sigma_{thres}^2 = 0.8$ mV$^2$ whereas the variance in synaptic potentials was more than 17–fold higher

($\sigma^2_{Vm} = 14.0$ mV$^2$). Therefore a randomness in the threshold had little of no effect on the irregularity of spiking compared with the randomness in synaptic input. In some recordings the threshold may appear as uncorrelated with the membrane potential prior to the spike onset. However, rather than a noisy threshold this is likely attributed to cellular morphology. If the cell is not electrically compact, the axon initial segment, where the spike is initiated, will have a different potential than what is recorded with the electrode. If this was the case, these observations would still be compatible with the two–regime hypothesis, since spikes would still be driven either by fluctuations or a large mean current, despite the disguise of a long electrotonic distance to the recording site.

## Rich diversity in population firing rates

So far the analysis has been performed on serially acquired intracellular recordings across trials and animals. This demonstrates that some neurons spiked primarily in the fluctuation–driven regime while others spiked in the mean–driven regime. Nevertheless, it is still unclear what the parallel population activity was during a behavior and across behaviors. How many neurons were in one versus the other regime and for how long? First, we assessed the neuronal participation in the motor patterns by their degree of spiking during motor behavior. Neurons were active during both ipsi– and contralateral scratching behaviors (*Figure 9A–D*). Most units had a rhythmic relationship with the nerve signals and a higher firing rate for the ipsilateral scratching compared with contralateral scratching behavior (cf. *Figure 9C and D*; *Videos 1* and *2*), which indicates participation of neurons in a hemicord to a smaller degree in the contralateral movement than the ipsilateral movement.

The distribution of firing rates across the neuronal population over several trials was strongly skewed, which indicate that most neurons spike relatively infrequently with a 'fat-tail' of higher spiking (*Figure 9E*). The distribution covered two orders of magnitudes from 0.1–10 Hz and was akin to a lognormal distribution (inset and green lines, *Figure 9E*). Similar lognormal–like distributions have been observed in other parts of the nervous system (*Buzsáki and Mizuseki, 2014*). The implication of the skewed distribution is that most neurons spiked at low rates, but there was relatively many neurons spiking at higher rates indicating an overall rich diversity of firing rates.

## Skewness preserved across behaviors

Although multi–functional spinal units have been reported previously (*Berkowitz et al., 2010*) it is unclear how their participation is distributed and whether the asymmetry in distribution is linked to different behaviors. To address this issue we analyzed the population spiking for multiple motor behaviors. The induction of a distinct scratch behavior is location–specific (*Stein, 2005*). Multiple behaviors can be evoked depending on exact location and which side of the body is touched. This allowed us to induce two distinct behaviors: ipsi– and contralateral hindlimb scratching, while recording from the same neuronal ensemble (*Videos 1* and *2*). These behaviors were reproducible over multiple trials (>9 trials). Both behaviors had similar phase relationships between the muscle synergists, although ipsilateral scratching had stronger activity (cf. *Figure 9A and B*). The firing rate distribution was positively skewed in both behaviors with the similar qualitative shape (*Figure 9E–F*). This skewness was also found across animals (green bars, *Figure 9G*, n=5) and close to zero on log–scale, i.e. lognormal (black lower bars). To further quantify the uneven neuronal participation we used Lorenz statistics and the Gini-coefficient (*O'Connor et al., 2010*; *Ikegaya et al., 2013*). The Lorenz curve characterizes the share of cumulative participation of individual neurons of the population (*Figure 9H*). The diagonal corresponds to the case where all neurons have the same firing rate. The deviation from equality is quantified by the Gini–coefficient, which is the fraction of area $a$ to the total area $a + b$ (*Figure 9H*). The higher the coefficient, the more unequal the participation across neurons is. Both scratch behaviors had a Gini–coefficient of ∼0.5 (*Figure 9I*). Although the mean firing rate could change between behaviors and between animals (*Figure 9J*), the skewness was qualitatively similar (*Figure 9K*). This suggests that the skewed lognormal–like firing rate distribution, and hence a presence of the fluctuation–driven regime, was preserved across behaviors and animals.

## Skewness in firing rate distribution is activity–dependent

Neurons do not occupy either the fluctuation– or the mean– driven regime all the time. Individual neurons can move back and forth between regimes depending on the synaptic current they receive. Neurons that spike predominantly in the mean–regime will have their mean firing rates closer

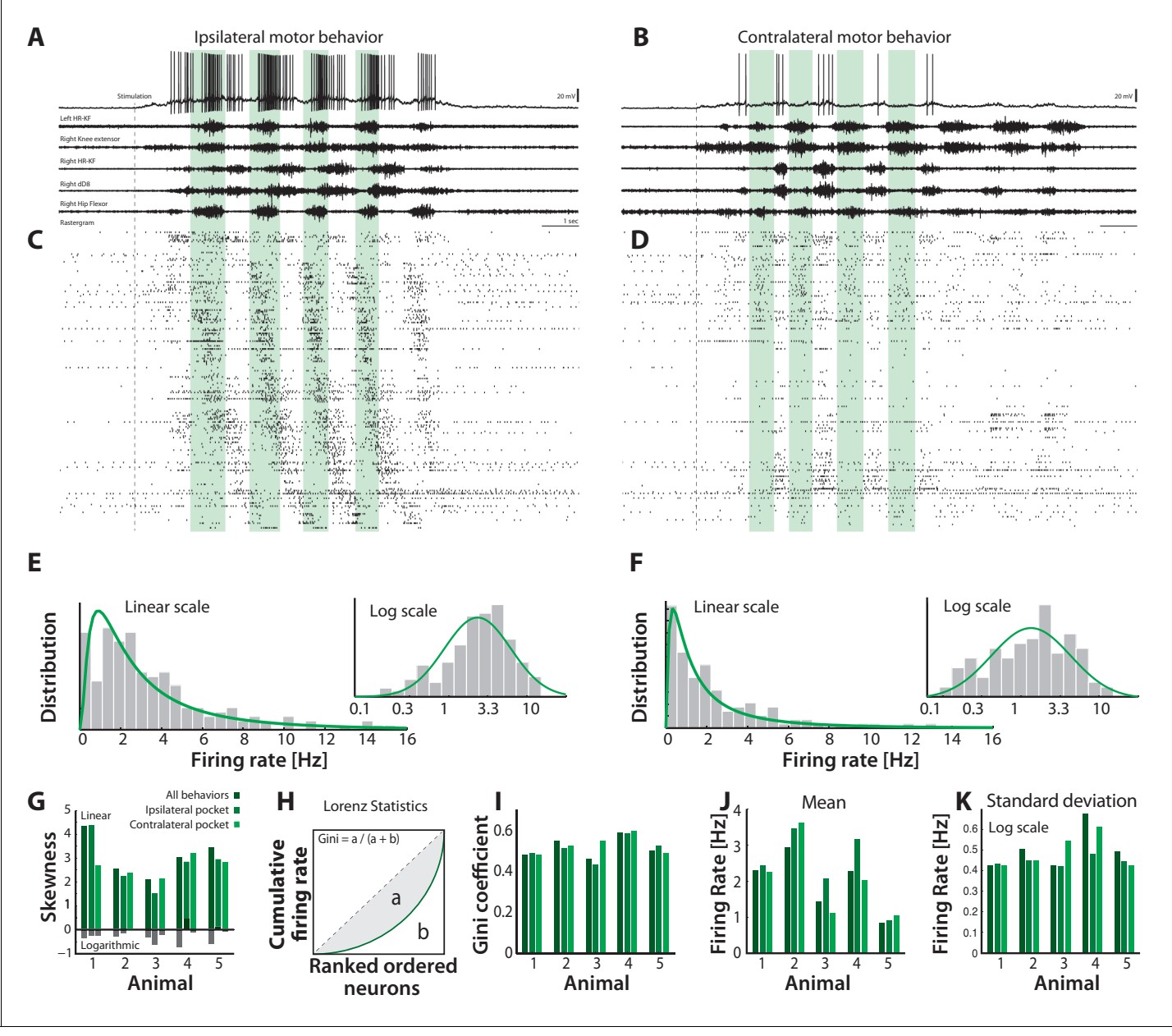

**Figure 9.** Skewed neuronal participation across behaviors. (**A–B**) Two distinct motor behaviors: Ipsilateral pocket scratch (left panel) and contralateral pocket scratch (right panel) shown by intracellular recordings (top) and motor nerve activities. (**C–D**) Rastergrams showing the unit activities during ipsilateral pocket scratch (**C**) and contralateral pocket scratch (**D**). Green areas mark the hip flexor phase. (**E–F**) spike count firing rate distribution for the behaviors on linear and a semi-log plot (insets), indicate lognormal participation. Lognormal functions are fitted (solid green lines). (**G**) Skewness on logarithmic (green bars) and linear scale (gray bars) is preserved across animals. (**H**) The inequal neuronal participation is calculated using Lorenz curve and gini coefficient. (**I**) Gini–coefficients cluster around 0.5 across behaviors and animals. Mean (**J**) and standard deviation on (**K**) of the distribution of firing rates on log–scale across behaviors and animals. $V_m$ resting level in (**A–B**) is −60 mV.

together and more normally distributed compared with those spiking in the fluctuation–regime. Hence, we expected the skewness of the distribution of mean firing rates across the population to become more negative (on log–scale) as the general network activity increases. To address this, we analyzed the spiking across neurons in parallel. First, we estimated the time–dependent firing rate of each neuron in the population using optimal Gaussian kernel (**Shimazaki and Shinomoto, 2010**) and measured skewness of the population distribution. The time–dependent population distribution was

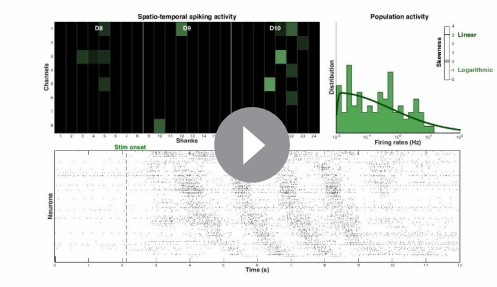

**Video 1.** Skewness of the population firing rate is activity–dependent: Behavior 1 (ipsilateral scratching). The spiking activity in three lumbar segments shown as a 24 by 8 pixel-grid, with each pixel representing a recording channel (top left panels, segments D8, D9 and D10 indicated). Columns represent probe shanks (separated by 200 μm) and rows the vertical positions in the dorsoventral axis (~30 μm between each). The light intensity of a pixel indicate the local firing rate in time estimated using Gaussian kernels. The time-dependent distribution of firing rates across the population (green histogram, top right, logarithmic x-axis) was fitted with a lognormal function (appearing here as a normal distribution) with variable skewness (solid black line). Skewness of fit on linear and log scale is shown on slider (inset). Note the dependence on overall activity. Lower panel: spike time rastergram (horizontal lines represent spiking of the neurons, which are sorted according to phase) and time is indicated with a black bar. The scratch reflex was activated at the time-point of the vertical dotted line ('Stim onset'). Sound is the aggregate spiking activity of the population.

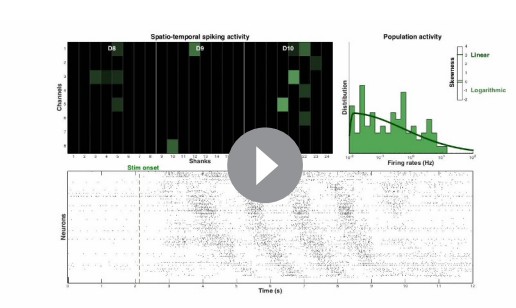

**Video 2.** Skewness of the population firing rate is less activity–dependent: Behavior 2 (contralateral scratching). Same neuronal activity as in *Video 1*, except the spinal network is now generating a different behavior. The neuronal ensemble spikes at a lower overall rate, which is reflected in a weaker relationship between skewness and activity (compare with *Video 1*).

achieved by binning the rates in 10 ms windows (*Videos 1* and *2*). The mean population rate and its SD are indicated as black $\pm$ gray lines (*Figure 10A*). As the mean firing rate increased, the skewness of the distribution (log–scale) became negative, which is a sign of more neurons were occupying the mean–driven regime (cf. inset histograms, *Figure 10A*). This was further confirmed by a negative correlation between the mean firing rate (black line in A) and the log–skewness for all time points (*Figure 10B*). Hence, as the general activity increased, the population distribution became less lognormal and more Gaussian, which suggests more neurons occupied the mean–driven regime during a higher general activity.

## Occupancy within regimes across population and time

To further gauge the division of neurons in the two regimes we compared the irregularity of the spiking using $CV_2$. This metric was verified above as a reliable indicator of spiking regimes. The distribution of the mean $CV_2$ across the population of neurons was clustered around 1 if all ISIs were included (gray histogram, *Figure 10C*). However, measuring the irregularity in the motor cycles alone i.e. excluding the inter–burst intervals (here, $ISI < 0.5$ s) the mean irregularity across neurons was lower and clustered around 0.6 (red histogram). Both distributions had substantial spread around the mean, which suggests a rich diversity spiking patterns.

To get a compound measure of the behavior of the entire population across time, we considered the amount of time each neuron spent in the fluctuation–driven regime. We demarcated the fluctuation–regime as having irregularity in spiking above a critical value, i.e. $CV_2 > i_{crit}$. Choosing $i_{crit}$ is not entirely objective. Complete Poisson–type irregularity has $CV_2 = 1$, but the spiking is still irregular for lower values (*Feng and Brown, 1999*). Based on our data, even when the $CV_2 \approx 0.5$, the $V_m$ spent as much as 96% of the time below threshold (*Figure 6C–D*) indicating fluctuation–driven spiking. Further, neurons that had $CV_2 \approx 0.5$, also had lognormal firing rate distributions (*Figure 7*), which also indicates the fluctuation–driven regime. For these reasons, we suggest choosing $i_{crit} = 0.5$ for distinguishing regular vs. irregular spiking. A similar value was previously chosen to distinguish between regular vs. irregular 'choppers' in the cochlear nucleus (*Young et al., 1988*). Thus, the population of spinal neurons had a large diversity in time spent in the fluctuation–driven regime. Some neurons spent as little as 20% in

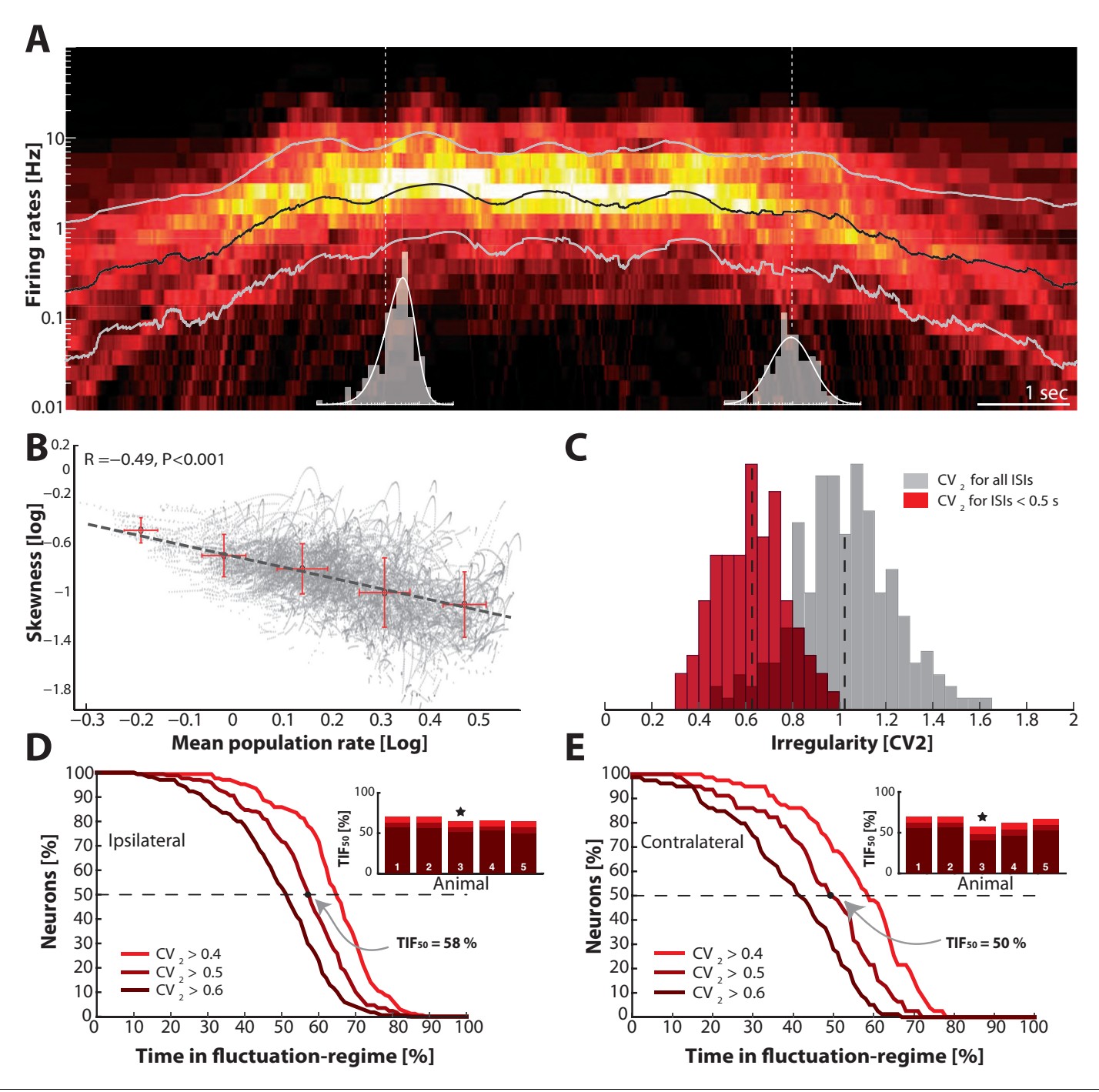

**Figure 10.** Skewness and irregularity across the neuronal population gauge occupation in both regimes across time. (A) Heat map of the distribution of firing rates across the population ($n = 190$ units, 1 animal) on log–scale (y–axis) as a function of time (x–axis). Lognormal mean ± SD are indicated as black and grey lines, respectively. Distribution is indicated (gray histograms) at two different time points (broken vertical lines). (B) Lognormal mean population firing rate (black line in A) versus log–skewness are negatively correlated, indicating more neurons move into mean–driven regime as the population rate increases. Scatter due to multiple trials, which is binned in sections, red crosses. (C) Distribution of irregularity (mean $CV_2$) across population for all ISIs (gray) and when excluding of inter–burst intervals (red). (D) Fraction of neurons, which spend a given amount of time in fluctuation–driven regime ($i_{crit} = 0.4, 0.5$ and $0.6$) normalized to 100% (Reverse cumulative distribution). The least time spent in fluctuation–driven regime by half of the neurons ($TIF_{50}$) is given by the intercept with the broken horizontal line and distribution (indicated by arrow). For this sample animal and behavior $TIF_{50} = 56\%$. Inset: Values across animals, sample animal indicated ($\star$). (E) The $TIF_{50}$–values across animals in both behaviors as indicated by similarity in values are remarkably conserved.

*Figure 10 continued on next page*

*Figure 10 continued*

The following figure supplement is available for figure 10:

**Figure supplement 1.** Distribution of neurons having fluctuation driven spikes and $SIF_{50}$ values.

the fluctuation–driven regime while other spent as much as 80%. To get a quantitative handle on the occupation of neurons in the fluctuation–driven regime across the population, we considered the distribution of time spent with $CV_2 > i_{crit}$. This was formally quantified using the reverse cumulative distribution of neurons that spend a given fraction of time in the fluctuation–driven regime (*Figure 10D*). The reverse cumulative distribution is plotted for 3 values of $i_{crit}$ (0.4, 0.5 and 0.6) to indicate the sensitivity to parameter choice. Obviously, choosing a lower $i_{crit}$ results in a larger fraction of time in the fluctuation–driven regime, i.e. the curve is shifted to the right. Choosing $i_{crit}$ larger has the opposite effect. This inverted *S*–shaped curve gives the fraction of neurons (y–axis), which spend at least a given time in the fluctuation–driven regime normalized to 100% (x–axis). Hence, half of the population spent at least 58% of time in the fluctuation regime during ipsilateral scratching (intercept of curve with the broken line, *Figure 10D*). We refer to this metric as the time in the fluctuation–regime for half the neurons ($TIF_{50}$). Similar $TIF_{50}$–values were obtained for all five animals (inset histogram). Qualitatively similar results were achieved for a different motor behavior, namely contralateral scratching (*Figure 10E*). The $TIF_{50}$ metric is a time–weighted analysis of irregularity of spike trains. In addition to measuring the time in regimes, we measured how many spikes were in one regime vs. the other. Hence, we calculated the reverse cumulative distribution of neurons that had a given fraction of spikes in the fluctuation–driven regime (*Figure 10—figure supplement 1*). Similar to $TIF_{50}$, we defined a spike–weighted metric as the *spikes in fluctuation regime for half the neurons* ($SIF_{50}$). Both the $SIF_{50}$– and $TIF_{50}$–values were relatively conserved across animals as well as behaviors (*Figure 10D–E*, *Figure 10—figure supplement 1*). The large values of $TIF_{50}$ and $SIF_{50}$ indicate that the fluctuation–driven regime had a strong presence during motor behaviors, and the high similarity suggests that it may represent a conserved fundamental property of network activity.

## Cell types and spiking activity

In the data analyses presented so far we have not addressed the neuronal identity of the recorded units. Nevertheless, there is a spatial division subtypes of spinal neurons, which we could take advantage of. During development, a distinct laminar organization of different cellular subtypes is formed in the dorsoventral axis (*Arber, 2012*; *Jessell, 2000*). In particular, motoneurons are primarily found in the most ventral part of the horn whereas interneurons are found in more medial to dorsal territory. Since this is the same axis that our electrode arrays were located along, it was possible to infer a likelihood of cellular identity based on location. The electrode shanks have multiple distributed electrodes (*Figure 11A*), which made it possible to approximate the soma location using *trilateration*. Trilateration is the geometrical process of determining the location of a source in space using multiple recording sites combined with the fact that signals decay in the extracellular space (*Manolakis, 1996*). Thus, the node locations were approximated based on the amplitude of spike waveforms, which clearly decayed with distance (*Figure 11B*). Node locations were combined for all shanks, probes and animals to form a scatter (*Figure 11C*). Combining these locations with depth of individual shanks with respect to the surface of the spinal cord, we were able to investigate the spike patterns with respect to the absolute neuronal location. The irregularity in spiking was quantified (mean $CV_2$) with respect to dorsoventral depth (*Figure 11D*). The distributions of mean firing rates (not shown) and the mean $CV_2$ (*Figure 11E*) had no obvious dependence on depth. In particular, the spread in means was much smaller than the SD of the distributions themselves. The most parsimonious interpretation of these data is that the fluctuation–driven spiking regime was both present and equally prominent in all the neurons, regardless of whether the cell body was in the ventral horn or in the medial horn, i.e. equally present in motoneurons and interneurons.

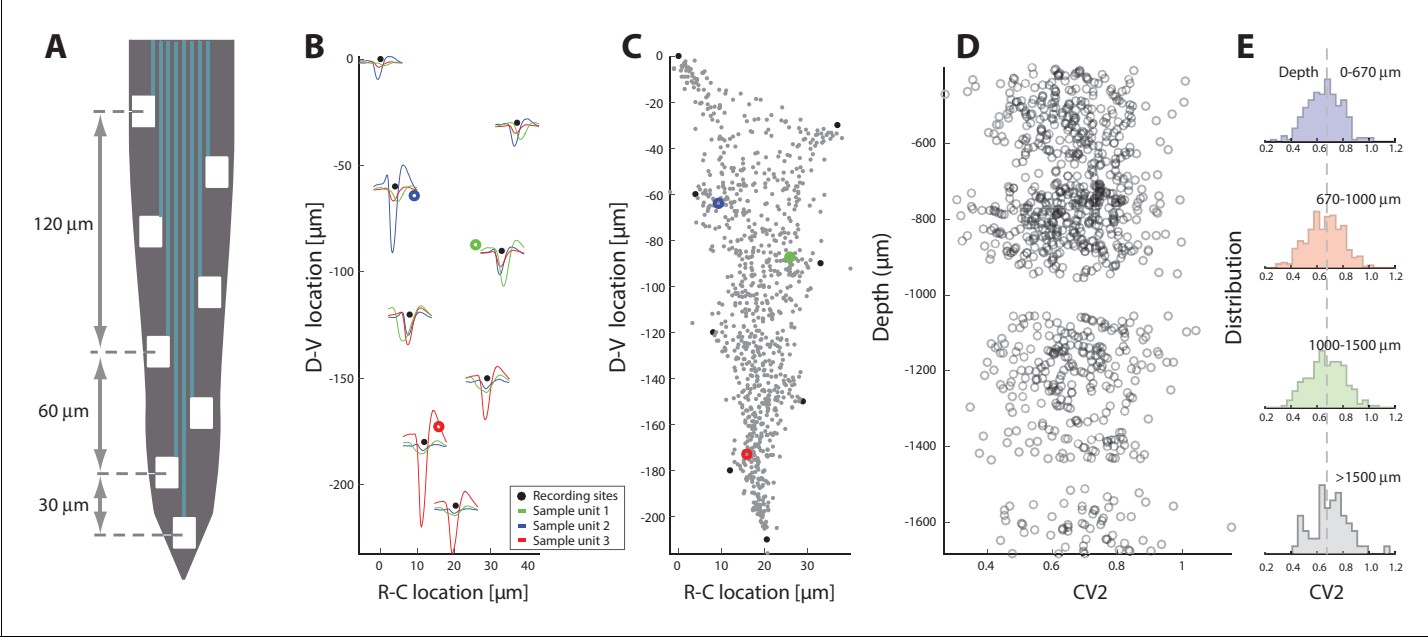

**Figure 11.** Spiking irregularity is independent of cellular location. (A) Layout of the 8 electrodes on a shank, which span a total of 210 μm in the dorsoventral axis. (B) Recorded waveforms at different locations of three sample units (colored in red, blue and green). The node locations are estimated via trilateration and indicated as rings. Electrode locations are indicated as black dots. (C) Composite of source-locations for all shanks and all animals (total $n = 921$ cells). The location of sample units from B indicated in colors. (D) Irregularity of the associated spiking are estimated (mean $CV_2$ on x-axis) versus the dorsoventral location (y–axis), where the unit locations are corrected for the depth of the individual shank with respect to the spinal cord surface. (E) The binned distributions of $CV_2$ as a function of depth. The distribution means are remarkably similar (broken line as fiducial) and a KS–test indicates no significant difference.

## Discussion

In neuronal networks, spikes are generated in either the *mean– or the fluctuation–driven regime* (*Brunel, 2000*; *Gerstner et al., 2014*; *Kuhn et al., 2004*; *Tiesinga et al., 2000*). In this report we present evidence for the existence of both regimes during motor pattern generation in the spinal cord. We consistently found normally distributed synaptic input combined with the supralinear shape of the IO–function in the subthreshold region, and suggest this as a compelling mechanism behind the lognormal population firing rate distribution (*Roxin et al., 2011*). Using spiking irregularity across the neuronal population as a hallmark of the fluctuation regime, we found that half of the neurons spent at least 50% of the time in this regime. Thus, the fluctuation–regime was not a rarity, but rather had a prominent presence both across behaviors and across animals (*Figure 10*). To our knowledge this is the first report, which quantifies occupation within spiking regimes of a neuronal population, not just in the spinal cord, but also in the nervous system in general.

### Stability and the two regimes

The fact that the relative time during which a subset of neurons occupied one of the two regimes was conserved across both behaviors and animals could indicate a key principle of neuronal processing. A fundamental challenge for neuronal networks is to perform functions while keeping the population activity from falling into either of the two extreme states: (1) the quiescent state where the neuronal spiking activity cannot remain self–sustained and (2) the unstable state of run–away recurrent spiking activity (*Vogels et al., 2005*; *Kumar et al., 2008*). It is well known that recurrent inhibition is important for maintaining stability, but other mechanisms may participate as well, e.g. synaptic depression or active adjustment of the shape of the neuronal response function by adaptation of spiking threshold. A nonlinear response function, as we observed in the fluctuation–regime (*Figure 4B*), will amplify input via supralinear summation (*Rubin et al., 2015*). The upward curvature will enhance synaptic fluctuations, which then accelerates the recurrent excitatory activity causing a

potentially unstable state. In contrast, the response function in the mean–driven regime, is linear or even sublinear, which is likely to curb strong input. We therefore propose that the close proximity of the $TIF_{50}$–value to 50% is an indication of a self–organizing trade–off between sensitivity and stability in order to preserve at once both network homeostasis and dynamical functionality. This conjecture remains to be further substantiated in future studies. Furthermore, the $TIF_{50}$– and $SIF_{50}$–values remain to be determined for other part of the nervous system and in other species.

## Rhythm generation and regimes

The distinction between fluctuation– and mean–driven spiking is interesting because the two types of spiking may have radically different causes, and this may hold an important clue to understanding the enigmatic motor rhythm generation. The fluctuation–driven spiking is believed to be caused by concurrent and random arrival of excitatory and inhibitory potentials resulting in a fluctuating sub–threshold $V_m$ (*Table 1*). In the mean–driven regime, on the other hand, the net membrane current is so large that the mean $V_m \pm \sigma$ is above threshold, and the ISIs are therefore determined by the recharging of the membrane capacitance following the refractory period of the previous spike (*Powers and Binder, 2000*). This results in a deterministic trajectory of $V_m$ and regular ISIs. More importantly, for the mean–driven spiking the membrane current can be caused by intrinsically electrical properties as well as synaptic input, whereas the fluctuation–driven spiking is exclusively caused by synaptic input. An intrinsic property, which is commonly believed to be involved in rhythm–generation, is the pacemaker property that can autonomously generate neuronal bursting in the absence of synaptic input (*Brocard et al., 2010*; *Ramirez et al., 2004*; *2011*). The prominent presence of the fluctuation–regime therefore implies that the majority of neuronal spikes were not driven primarily by intrinsic properties such as pacemaker potentials, but rather by synaptic communication. This can be interpreted in two ways: (1) if there is a pacemaker–driven rhythmogenic core of oscillatory neurons responsible for the motor activity (*Huckstepp et al., 2016*), the core only represents a small fraction of the network, or (2) since the fluctuation–regime is prominent and pacemaker neurons are difficult to find, the motor–rhythm may be generated by other means such as emergent collective processes in the network (*Yuste, 2015*). Generation of movements without the need of pacemaker neurons have been predicted theoretically in central pattern generators (*Kleinfeld and Sompolinsky, 1988*) as well as more complex sequence generation (*Hennequin et al., 2014*). Even in the respiratory system, which has the most stereotypic motor rhythm, pacemaker cells appear not to be essential for generation of the rhythmic breathing, although this topic is still debated (*Feldman et al., 2013*; *Ramirez et al., 2011*; *Carroll and Ramirez, 2013*; *Chevalier et al., 2016*). It remains to be understood how a distributed emergent processes can generate motor rhythms on a network level if, in fact, the pacemaker bursting is not an essential component.

## Cell identity and circuit function

In spinal research, neuronal identification has improved over the last decades with the development of genetic knockouts and molecular markers (*Bikoff et al., 2016*; *Goulding, 2009*; *Britz et al., 2015*; *Kiehn, 2006*). Pinning down cellular identity improves the search for a potential specialization in the circuit. However, the sole focus on cellular identity to address questions in spinal research carries a weakness as well as a strength. It contains the risk of missing the collective dynamics and the delicate interaction among neuronal cell types. Neural circuits operate to perform functions by collective interaction between all neurons, where it is difficult, if not impossible, to link a particular function to the individual neuron. Functional activity may very well arise on circuit level as opposed to cellular level. This caveat is known as the neuron doctrine versus emergent network phenomena (*Yuste, 2015*; *Grillner, 2006*), and the neuron doctrine has almost exclusively been adopted in previous investigations of spinal motor circuits. To the best of our knowledge this report is the first investigation of spinal motor circuits from an ensemble viewpoint.

Nevertheless, since motoneurons are fundamentally different from the rest of spinal neurons it would be helpful to distinguish them from interneurons. In our experiments we sampled from neurons, which were likely to be primarily interneurons since they are more numerous than motoneurons. The fraction of motoneurons to interneurons is 1:8 (*Walloe et al., 2011*), but we were also likely to sample motorneurons, since they have large somata. To explore this further, we investigated the population activity and its relation to cellular identity by taking advantage of their spatial

segregation in the dorsoventral axis (*Arber, 2012*; *Jessell, 2000*). We were able to associate an absolute location of the cellular somata (using trilateration), and thus test for differences in spiking activity (*Figure 11*). The distribution of firing rates as well as the spiking irregularity did not have any dependence on location. This suggests that the fluctuation–driven spiking regime was both present and equally prominent in all the neurons, regardless of whether the cell bodies were in the ventral or medial horn, i.e. regardless of whether they were motoneurons or premotor interneurons.

## Comparison with other parts of the CNS

Common features of network activity for different parts of the central nervous system may provide hints towards fundamental principles of neuronal operations. In the present study we identified the following features of population motor activity: (1) synaptic input to individual spinal neurons was normally distributed (*Figure 3*), (2) the means of these normal distributions were also normally distributed across the population. In particular, the distance to threshold in terms of fluctuations, i.e. $(V_m - V_{thres})/\sigma$ had a normal distribution and a distance from mean to threshold of $3\sigma$ on average (*Figure 3—figure supplement 2F*). (3) The neuronal response function was supralinear when the mean input was in the subthreshold region (*Figure 4*). (4) There was a rich diversity of regular to irregular spiking patterns. (5) The population firing rate was skewed and lognormal–like.

Many of these features have been identified before in other parts of CNS. The $V_m$ of individual neurons is often normally distributed in cortical neurons when considering either the up– or down–state (*Destexhe et al., 2003*; *Stern et al., 1997*) and the spiking is irregular with a $CV$ clustered around 1 (*Softky and Koch, 1993*; *Stevens and Zador, 1998*). Similar irregularity is observed in invertebrates (*Bruno et al., 2015*). The distribution of mean $CV_2$ values in our experiments was clustered around 0.6 when ignoring the inter–burst intervals (*Figure 10C*). This is more regular than what is observed for typical cortical neurons (although see *Feng and Brown, 1999*), but similar to cervical interneurons in monkeys performing isometric wrist flexion–extensions (*Prut and Perlmutter, 2003*).

## Lognormal population firing

We observed a skewed and lognormal–like population distribution across behaviors (*Figure 9*, *Videos 1* and *2*). Similar lognormal distributions have been reported in other parts of CNS (*Buzsáki and Mizuseki, 2014*; *Hromádka et al., 2008*; *O'Connor et al., 2010*; *Wohrer et al., 2013*) and it remains an open question how the skewness arises out of neuronal ensembles. Roxin et al proposed the mechanism where the skewness arises from a nonlinear transformation of Gaussian input (*Roxin et al., 2011*). Our data supports this hypothesis. First, we observed a normally distributed $V_m$ for individual cells, which is a proxy for the requirement of normally distributed input currents (*Figure 3*). Second, a supralinear IO–function covering most of this input (*Figure 4*). Third, a firing rate distribution of individual cells which was typically highly skewed and lognormal–like although some did not have lognormal firing (*Figure 5*). Nevertheless, there is a distinction between the lognormal firing of individual neurons and the lognormal distribution of mean rates across the population. Whereas the lognormal population firing rate remains to be fully understood, the skewed firing rate distribution of individual neurons is fairly well understood. Here, the skewness is due to the fluctuating input and irregularity of spiking (*Ostojic, 2011*). Nevertheless, we argue the mechanism for the lognormal population firing is the same as that for the individual neuron. If the subthreshold IO-function is approximately similar across the population, which our data implies (*Figure 4*), we can explain the lognormal population firing by a supralinear transformation, if the mean $V_m$ across the population is also Gaussian. We did in fact find the distribution of mean $V_m$ to be Gaussian (*Figure 3—figure supplement 2F*).

## Fluctuation–driven regime as a subprimary range in motoneurons?

Classical studies of spinal motoneurons indicate two regimes of spiking: a primary and a secondary range (*Kernell, 2006*; *Meehan et al., 2010*), which corresponds to different parts of the mean–driven spiking regime. This characterization was associated with the intrinsic properties without synaptic input being present. Nevertheless, a different type of fluctuation–driven spiking was discovered in experiments where synaptic input were present, in what was referred to the subprimary range in mice (*Manuel and Heckman, 2011*) and humans (*Kudina, 1999*; *Matthews, 1996*). This subprimary

range conforms to the fluctuation–regime though under a different terminology. As the name indicates, the primary range has been considered to represent the dominant mode of spiking whereas the subprimary range is a peculiarity. Nevertheless, a recent study recorded for the first time the motoneuron discharge and muscle force and found that the subprimary range accounts for 90% of the contraction force (*Manuel and Heckman, 2011*). This indicates that the fluctuation–regime may have a more noteworthy role in covering the dynamical range in motor control than previously assumed, which is in agreement with the observations of the present study.

## Materials and methods

### Experimental procedures

The experimental procedures are described in more detail at Bio-protocol (*Petersen and Berg, 2017*). We used the integrated turtle preparation with the spinal motor network intact ($n = 5$ for the multi–electrode recordings and $n = 60$ for the serially aqquired intracellular recordings), in order to address how the neuronal firing rates are distributed across the population of interneurons and motoneurons in the spinal cord (*Petersen et al., 2014*). These sample sizes where assumed to be large enough in the experimental design and because of a consistency in results, although a specific power analysis was not conducted. To avoid the confounding factors of supraspinal input, we spinalized the turtle. The transection was performed at the spinal cord at segments (D3-4) caudal to the cervical segments, where the local circuitry has only little or no involvement in generation of motor patterns (*Mortin and Stein, 1989*; *Hao et al., 2014*; *Mui et al., 2012*). The adult turtle preparation is capable of producing elaborate motor patterns such as scratching. We used the semi-intact spinal cord of adult turtles (*Keifer and Stein, 1983*; *Petersen et al., 2014*) and recorded from the segments D8-D10. These segments contain the essential CPG circuits (*Mortin and Stein, 1989*). Most of the spinal cord including the sensory periphery is left intact. The blood is replaced and the spinal column is provided with oxygenated Ringer's solution so that the neurons and the network have optimal conditions. In this experimental situation the motor behavior is as close to in vivo situation as possible, and is indistinguishable from the intact condition (*Keifer and Stein, 1983*). The turtle preparation allow for mechanical stability and the turtle's resistance to anoxia allow for remarkable durability of both the recording conditions and the motor pattern reproducibility (*Vestergaard and Berg, 2015*).

### Integrated preparation

Adult red-eared turtles (*Trachemys scripta elegans*) of either sex were placed on crushed ice for 2 hr to ensure hypothermic anesthesia. The turtles were killed by decapitation and the blood was substituted by the perfusion with a Ringer's solution containing (mM): 100 NaCl; 5 KCl; 30 NaHCO$_3$; 2MgCl$_2$; 3CaCl$_2$; and 10 glucose, saturated with 95% O$_2$ and 5% CO$_2$ to obtain pH 7.6, to remove the blood from the nervous system. We isolated the carapace containing the spinal cord segments D4-Ca2 by transverse cuts (*Keifer and Stein, 1983*; *Petersen et al., 2014*) and perfused the cord with Ringer's solution through the vertebral foramen , using a steel tube and gasket pressing against the D4 vertebra. We opened the spinal column on the ventral side along D8-D10 and gently removed the dura mater with a fine scalpel and forceps. For each insertion site for the silicon probed, we opened the pia mater with longitudinal cuts along the spinal cord with the tip of a bend syringe needle tip (BD Microlance 3: 27G3/4", 0.4x 19 mm). We performed the cuts parallel to the ventral horn between the ventral roots. The surgical procedures comply with Danish legislation and were approved by the controlling body under the Ministry of Justice.

### Network activation

We used a fire polished tip of a bent glass rod for mechanical stimulation, that was mounted linear actuator. The actuator was controlled with a function generator: frequency, amplitude and duration of the stimulus.

### Extracellular recordings

Extracellular recordings were performed in parallel at 40 KHz using a 256 channel multiplexed Amplipex amplifier (KJE-1001, Amplipex). Up to four 64-channel silicon probes were inserted in the

incisions perpendicular to the spinal cord from the ventral side. We used the 64-channel Berg silicon probes (Berg64 from NeuroNexus Inc., Ann Arbor, MI, USA) with 8 shanks, and 8 recording sites on each shank arranged in a staggered configuration with 30 μm vertical distance. The shanks are distanced 200 μm apart. Recordings were performed at depths in the range of 400-1000 μm.

## Intracellular recordings

The intracellular recordings were performed in current-clamp mode with an Axon Multiclamp 700B amplifier (Molecular devices). Glass pipettes were pulled with a P-1000 puller (Sutter instruments) and filled with a mixture of 0.9 M potassium acetate and 0.1 M KCl. Data were sampled at about 20 kHz with a 12-bit analog-to-digital converter (Axon Digidata 1440a, Molecular devices). We inserted the glass electrodes from the ventral side of D8-D10 perpendicularly to the spinal cord. Neurons were located at depths ranging from about 300–800 μm. Typically we had stable intracellular recordings for multiple trials.

## Nerve recordings

Electroneurogram (ENG) recordings were performed with suction electrodes. The scratch behavior was measured by the activity of the nerves: Hip Flexor, Knee Extensor, dD8 and HR-KF. The nerve activities were recorded with a differential amplifier Iso-DAM8 (World Precision Instruments) with bandwidth of 100 Hz–1 kHz.

## Histology

For histological verification, we combined several staining techniques: The silicon probes were painted with DiI (1–2% diluted in ethanol) before insertion into the spinal cord (*Blanche et al., 2005*; *Vandecasteele et al., 2011*). Following successful experiments, we performed Nissl– and ChAT– staining of the tissue, to determine the location of respectively neurons and motoneurons.

The histological processing is detailed in (*Petersen et al., 2014*). We carefully removed the tissue, perfused it and put it in phosphate buffered saline (PBS) with 4% paraformaldehyde for 24–48 hrs and further stored it in PBS. The tissue was mounted in an agar solution and sliced into 100 μm slices using a microtome (Leica, VT1000 S). The slices were washed with PBS and incubated overnight at 5°C with primary choline acetyltransferase antibodies goat anti-ChAT antibodies (1:500, Milipore, USA) in blocking buffer, which is PBS with 5% donkey serum and 0.3% Triton X-100. The slices were washed three times with PBS and incubated for 1 hr at room temperature with the secondary antibody Alexa488 conjugated to donkey anti-goat antibodies (1:1000 Jackson) in blocking buffer. After three washes with PBS, the slice was mounted on cover slit with a drop of ProLong Gold antifade reagent (Invitrogen Molecular Probes, USA) and cured overnight at room temperature before microscopy. The slice was viewed using a confocal microscope, Zeiss LSM 700 with diode lasers, on a Zeiss Axiolmager M2 using 10x/0.30 EC Plan-Neofluar, 40x/0.6 Corr LD Plan-Neofluar, and 63x/1.40 oil DIC Plan-Apochromat objectives (Zeiss).

## Data analysis

The data analysis was primarily done in the programming languages Matlab and Python. The correlation coefficient was calculated as the Pearson product-moment correlation coefficient.

## Skewness of distribution

We use *skewness* (*Press et al., 1992*) or the third moment as a measure of asymmetry in the distribution around the mean, sometimes referred to as Pearson's moment coefficient of skewness. It can be estimated using the method of moment estimator as

$$Skewness = \frac{1}{N}\sum_{j=1}^{N}\left[\frac{x_j - \bar{x}}{\sigma}\right]^3$$

where $x_1, ..., x_N$ are all the observations ($V_m$ or firing rate) and $\sigma$ and $\bar{x}$ are the sample standard deviation and sample mean of the distribution. The skewness is a unitless number and a value of zero indicates perfect symmetry. A positive skew has a tale pointing in the positive direction of the axis and a negative value points in the opposite direction.

## Spike sorting

Spike sorting was performed in the Klustakwik-suite: SpikeDetekt, KlusterKwik v.3.0 and KlustaViewa (*Kadir et al., 2014*). Raw extracellular signals were bandpass filtered from 400–9000 Hz, and spikes were detected by a median based amplitude threshold with SpikeDetekt (*Takekawa et al., 2012*; *Kadir et al., 2014*; *Quiroga et al., 2004*). An automatic clustering of the spikes was performed in KlustaKwik, followed by manual cluster-cutting and cluster verification in KlustaViewa. The cluster quality was evaluated by several measures: The shape of the autocorrelation function, the amount of contamination in the refractory period, the Isolation distance (*Harris et al., 2001*) and the $L_{ratio}$ (*Schmitzer-Torbert and Redish, 2004*) (*Figure 2—figure supplement 2*). Only well isolated units was used in the further data analysis.

## Time-dependent firing rates

The time-dependent firing rate $\nu$ was estimated by a gaussian kernel by convolving the spike times, $s(t)$, with a Gaussian kernel $k(t)$:

$$\nu(t) = \int_{-\infty}^{\infty} s(t - t')k(t')dt'$$

where $k(t)$ is defined as

$$k(t) = \frac{1}{\sqrt{2\pi}\omega} e^{-\frac{t^2}{2\omega^2}}$$

with the bandwidth $\omega$ optimized for each spike train with the sskernel method (*Shimazaki and Shinomoto, 2010*). The estimated width was in the range of 100–500 μs.

## Gini coefficient

The *Gini* coefficient is a measure of statistical dispersion and it is defined as a ratio of the areas on the Lorenz curve diagram

$$Gini = \frac{a}{a + b} = 1 - 2b$$

where $a + b$ is the area below the line of no dispersion (the diagonal, i.e. $a + b = 1/2$), and $b$ is the Lorenz curve, i.e. the cumulative distribution of firing rates (*Figure 9H*).

## Irregularity of the spiking activity

The irregularity of the spiking of individual neurons can be described by several measures. The most common measures are the coefficient of variation ($CV = \sigma/\mu$) and the Fano factor ($F = \sigma^2/\mu$), but both measures easily overestimate the irregularity when the firing rate is non-stationary (*Holt et al., 1996*; *Ponce-Alvarez et al., 2010*; *Softky and Koch, 1993*). More advanced methods of estimating the time dependent variations in the irregularity have been developed (*Shinomoto et al., 2009*; *Shimokawa and Shinomoto, 2009*; *Miura et al., 2006*), and here we use the widely used metric $CV_2$, which has been suggested to be the most robust measure of local spiking irregularity (*Wohrer et al., 2013*; *Ponce-Alvarez et al., 2010*). The time dependent $CV_2$ is defined by pairs of adjacent inter-spike intervals $ISI_i$ and $ISI_{i+1}$:

$$CV_2(i) = \frac{2|ISI_i - ISI_{i+1}|}{ISI_i + ISI_{i+1}}$$

where $CV_2 = 1$ for a Poisson process and $CV_2 = 0$ for a regular process. $CV_2$ can take values in the range from zero to two.

We noticed a small difference in the distribution of irregularity among the neurons recorded with intracellular versus extracellular electrodes (data not shown). The neurons were recorded with intracellular electrodes had more regular spiking than those recorded with extracellular electrodes. This may be caused by a systematic bias in the way the intracellularly recorded neurons were collected, as there is an experimental bias towards high firing rates. Spike sorting processing of the

extracellular recordings, on the other hand, is likely to both miss spikes and contain false positives, which will cause overestimation of spiking irregularity.

## $TIF_{50}$ and $SIF_{50}$: time and spikes in fluctuation regime based on spiking irregularity

To get a quantitative handle on the fraction of neurons found in the fluctuation–regime across the population, we consider the distribution of neurons, $f(t)$, which spends a given amount of normalized time $t$ in the fluctuation regime, i.e. with $CV_2 > i_{crit}$. We consider three values of $i_{crit}$, 0.4, 0.5 and 0.6, as indicators for when the neurons are in the fluctuation–regime. Formally we quantify the time in fluctuation–regime for the population using the reverse cumulative distribution of neurons (*Figure 10D–E* and *Figure 10—figure supplement 1*). The reverse cumulative fraction of neurons in the fluctuation regime $F(t)$ for a given fraction of normalized time $t$ is

$$F(t) = 1 - \int_0^t f(t)dt, \ 0 < t \le 1$$

This fraction $F(t)$ is the fraction of neurons, which spend at least $t$ amount of normalized time in the fluctuation regime. To compress the distribution into a single number we use the fraction of time in fluctuation regime of half of the population, $TIF_{50}$, which is the value of $t$ for which $F(t) = 50\%$ (arrows and broken lines, *Figure 10D–E*).

Since the firing rate is rarely constant, one may want to know how many spikes are elicited in the mean– versus fluctuation regime. This is calculated in similar way, using the distribution of neurons having a normalized fraction of spikes in the fluctuation regime, i.e. spikes with $CV_2 > i_{crit}$, $f(s)$. The reverse cumulative of $f(s)$ again gives the fraction of neurons which have at least $s$ spikes in fluctuation regime, normalized to 100%,

$$F(s) = 1 - \int_0^s f(s)dt, \ 0 < s \le 1$$

Again we compress the distribution into a single number and use the fraction of spikes, which occur in fluctuation regime of half of the population, $SIF_{50}$, which is the value of $s$ for which $F(s) = 50\%$ (arrows and broken lines *Figure 10—figure supplement 1*).

## Estimating threshold

We use a definition of the action potential threshold, which is based on the phase plot of $V_m$ versus the derivative $dV_m/dt$. This is the second method reported in *Sekerli et al. (2004)*. The threshold is found as the point in the trajectory in phase space, where there is a strong departure from rest prior to the cycle. Since $dV_m/dt$ is proportional to the membrane current, this point represents a strong initiation of the inward current. Defining the slope of $V_m$ in time, $f = \frac{dV_m}{dt}$, the threshold is defined as the largest peak in second derivative with respect to $V_m$ in phase space, i.e. the maximum of $\frac{d^2 f}{dV_m^2}$ (red dots, *Figure 6—figure supplement 1B–C*). This is the point with the largest acceleration from baseline prior to the peak of the action potential. The $V_m$ trace was low–pass filtering at 5000 Hz to reduce the vulnerable to electrical noise of the estimates of derivatives.

## Spike rate versus $V_m$ (FV-curve)

The method for estimating the response rate as a function of $V_m$ has been described previously (*Vestergaard and Berg, 2015*). The relationship between firing rate, $\nu$, and membrane depolarization is based on the assumption that spikes occur as a random renewal point–process i.e. a Poisson process. The rate is directly related to the probability, $P$, of a spike occurring in a small time window at a certain time $t$:

$$P(t; t + \Delta t) = \nu \Delta t$$

The window $\Delta t$ has to be small such that the chance of getting more than one spike in the window is negligeble. The firing rate can thus be defined in terms of the probability of achieving a spike in an infinitesimally small time window (*Gerstner et al., 2014*):

$$\nu(t) = \lim_{\Delta t \to 0} \frac{P(t; t + \Delta t)}{\Delta t}$$

This definition of $\nu$ is also called the 'stochastic intensity'. Since the probability $P$ is strongly dependent on the depolarization of the membrane potential, the firing rate will be similarly dependent. To determine $\nu$ as a function of $V_m$ we have to empirically determine the probability, $P$, for the smallest possible value of $\Delta t$, which is the sampling interval of the intracellular recordings. To get $P$ as a function of membrane potential, $P(V_m)$, we first empirically determine the stochastic distribution of $V_m$ prior to the spike (1.5-1.7 ms prior), which we know will cause a spike. Then we normalize this distribution with the amount of time spent at each $V_m$-level at all time. This is the estimated probability of getting a spike, $P$, within a small time window $\Delta t$ for a given $V_m$, i.e. the firing rate as a function of $V_m$. This empirical method of relating firing rate and $V_m$ was relatively recently invented (*Jahn et al., 2011*) and used in determining IO properties of e.g. motoneurons (*Vestergaard and Berg, 2015*). The shape of the spike response function is highly non-linear with upward curvature. This has been observed in previous experiments (using a different method) and has often been referred to as expansive non-linearity (*Hansel and van Vreeswijk, 2002*; *Miller and Troyer, 2002*; *Murphy and Miller, 2003*; *Priebe and Ferster, 2005*, *2008*). An exponential

$$\nu(V_m) = c e^{\beta V_m}$$

was fitted to capture the curvature, where the curvature is represented in the exponent $\beta$, which have units of $1/mV$, and $c$ is a constant of units $1/s$. Such expansive non-linearities have also been investigated in the visual cortex where they are often characterized as a power-law relationship, i.e.

$$\nu(V_m) = k[V_m - E_a]^\alpha$$

where $k$ is a constant and $\alpha$ is the power >1, i.e. supralinear, and often ranging from 2-5 (*Hansel and van Vreeswijk, 2002*; *Miller and Troyer, 2002*). This exponent is also a measure of the expansive curvature of the non-linearity. $E_a$ represent a subthreshold level of $V_m$, where the spiking probability is zero, such that the values in the sampled traces are always larger than $E_a$, i.e. $V_m > E_a$. The curvature dependence on synaptic fluctuations was assessed by the standard deviation of the distribution of $V_m$ traces prior to the spike in the diffusion regime, i.e. where there was no link to the $V_m$ and the spike occurrence. This distribution was chosen 18 ms prior to the spike (*Figure 3B*). The analysis and fits were performed in Matlab with generic fitting functions.

## Return map ratio: Intracellular metric for mean– vs. fluctuation–regime

In order to distinguish neurons in fluctuation– versus mean–regime, we employ a new metric for quantifying the degree of fluctuations in $V_m$ in between action potentials. We plot the values of $V_m$ in a return map, which is a plot of $V_m(t)$ versus $V_m(t + \Delta t)$. If the inter–spike $V_m$ has a direct trajectory from the reset potential to the next spike, $V_m$ will smoothly increase and thus $V_m(t + \Delta t)$ will always be larger than $V_m(t)$. Therefore each point will be above the line of unity (*Figure 3—figure supplement 1A–B*). On the other hand, if $V_m$ has fluctuations, it will have an indirect and convoled trajectory from the reset value to the threshold. This will manifest as containing values of $V_m(t + \Delta t)$ which are actually smaller than $V_m(t)$. Thus we use the ratio of points above versus below the unity line as a metric for how convoled and fluctuating the path of $V_m$ is from reset to threshold. If the ratio is ~0.5 then $V_m$ is highly fluctuating, whereas if the ratio is approaching 1 the path is straight without any fluctuations. We choose a mean value of the histogram of all values to 0.7 to classify neurons as fluctuation– or mean–driven (*Figure 3—figure supplement 1C*). This metric of straight versus convoled trajectory had significant negative correlation with other measures of fluctuation– regime, e.g. spike rate skewness, spike irregularity ($CV_2$) and least time below threshold (LTBT, *Figure 3—figure supplement 1D–F*). The choice of $\Delta t$ is not important as long as it is larger than the timescale of electronic fluctuations of the amplifiers and smaller than the timescale of synaptic fluctuations in $V_m$. We consistently used $\Delta t = 1.5$ ms for all neurons. The return map ratio is intended as a metric to analyze sub-threshold activity and therefore spikes were removed from the traces, including a 6 ms window before and after the peak. Also, the $V_m$ containing the interburst (defined as having ISIs > 300 ms) intervals was removed.

## Determining cellular location using trilateration

Trilateration is a geometrical process of determining the location of a source in 2D–space using multiple recording sites scattered in space. We adapted the method to take advantage of a distance–dependent decay of the electrical signal from the action potential in the extracellular space. In this way, the amplitudes of the waveforms, which were simultaneously recorded on multiple electrodes, revealed the location of the source in space relative to the position of the electrodes. We assumed that the electrical signal decayed as $1/r^2$, where $r$ is the distance.

## Data selection

In *Figure 2*, the following trials were used: $n = [6, 4, 9, 5, 6]$ for ipsilateral pocket scratch and $n = [6, 3, 10, 5, 6]$ for contralateral pocket scratch. Data used in *Figure 7* has already been published in a different context (*Berg et al., 2007*). A small subset of the neurons used in *Figure 3D–E* ($n = 10$ out of 68) has been acquired in a reduced preparation (*Petersen et al., 2014*) and published for an investigation of a different matter (*Berg et al., 2007*; *Berg and Ditlevsen, 2013*). The data from experiments of blockade of inhibition using superfusion of strychnine has also been published previously in the investigation of a different matter (*Vestergaard and Berg, 2015*). Regarding excluding spikes from the analysis in *Figure 3C–E*: For the temporal distribution (panel C), only ISIs > 6 ms was included and for the spike triggered $V_m$-distribution only ISIs > 20 ms was included, all having ISIs < 300 ms. Estimating the FV-curve (*Figure 4*) all spikes having ISIs > 1.7 ms was included.

## Definition of fluctuation– and mean–driven spiking

Neuronal spiking has traditionally been considered to occur when the mean inward current of the cellular membrane is large enough to cross the rheobase such that the mean membrane potential ($V_m$) is above threshold ($V_{thres}$). In practice, the mean $V_m$ will not exceed $V_{thres}$ by very much due to the active spiking and after–hyperpolarization, but if this mechanism was turned off the mean membrane current ($I_m$) would drive $V_m$ across threshold, formally written as $I_m thres/R_m$ where $R_m$ is the membrane resistance. Spikes elicited in this manner are in the mean–driven regime (*Gerstner et al., 2014*; *Renart et al., 2007*). They have shorter inter–spike intervals (ISIs) because of the large $I_m$ and regular spiking due to the after–hyperpolarization. In contrast, when the mean $V_m$ is below threshold, i.e. $I_m < V_{thres}/R_m$, spikes are elicited by temporary fluctuations in $V_m$ due to synaptic bombardment. Such spiking is in the fluctuation–driven regime (*Kuhn et al., 2004*; *Tiesinga et al., 2000*; *Gerstner et al., 2014*; *Roxin et al., 2011*). The random synaptic fluctuations cause the spiking to be more irregular, which results in a higher coefficient of variation (CV, defined as the standard deviation ($\sigma$) divided by the mean of ISIs), than for the mean–driven regime (cf. *Figure 1D–E*). Therefore irregularity is an indicator of the spiking regime. Another indicator of the fluctuation–driven regime is positive skewness of the firing rate distribution (*Figure 1A–B*). These indicators are used to quantify the fraction of the population that is in one versus the other regime.

## Acknowledgements

Thanks to Gyӧrgy Buzsáki, Daniel F English and Henrik Lindén for reading and commenting on an earlier version of the manuscript. Funded by the Novo Nordisk Foundation (RB), the Danish Council for Independent Research Medical Sciences (RB and PP) and the Dynamical Systems Interdisciplinary Network, University of Copenhagen.

## Additional information

### Funding

| Funder | Author |
| --- | --- |
| Sundhed og Sygdom, Det Frie Forskningsråd | Rune W Berg |
| Novo Nordisk | Rune W Berg |

The funders had no role in study design, data collection and interpretation, or the decision to submit the work for publication.

## Author contributions
PCP, Conception and design, Acquisition of data, Analysis and interpretation of data, Contributed unpublished essential data or reagents; RWB, Conception and design, Acquisition of data, Analysis and interpretation of data, Drafting or revising the article, Contributed unpublished essential data or reagents

## Author ORCIDs
Peter C Petersen, http://orcid.org/0000-0002-2092-4791
Rune W Berg, http://orcid.org/0000-0001-6376-9368

## Ethics
Animal experimentation: The surgical procedures comply with Danish legislation and were approved by the controlling body under the Ministry of Justice.

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
