## [Decision Letter]

Thank you for submitting your article "Lognormal firing rate distribution reveals prominent fluctuation-driven regime in spinal motor networks" for consideration by *eLife*. Your article has been reviewed by three peer reviewers, one of whom, Jan-Marino Ramirez (Reviewer #1), is a member of our Board of Reviewing Editors and the evaluation has been overseen by Eve Marder as the Senior Editor. The other two reviewers involved in the review of your submission have agreed to reveal their identity: Mark Humphries (Reviewer #2) and Alexander Roxin (Reviewer #3).

The reviewers have discussed the reviews with one another and the Reviewing Editor has drafted this decision to help you prepare a revised submission.

Summary:

This study characterizes the firing rate distribution of neurons within spinal motor networks. The authors have beautifully combined intracellular and large-scale extracellular recordings to reveal the wide-spread existence of "fluctuation-driven" activity in spinal motor networks. These neurons (50% or more) discharge in a "subthreshold" manner. The authors suggest that the existence of these neurons is a strategy to increase stability and sensitivity in the locomotor network. The return map ratio as used here is a metric that has the potential for wide application in a variety of networks. Indeed, the authors nicely discuss their work in the context of a variety of networks. The study elegantly links spiking dynamics to subthreshold activity and therefore shows that the fluctuation-driven versus mean-driven regimes as described in theoretical work appears to be consistent with real networks. The strength of the study is the use of electrophysiology to test a theoretical framework on neuronal dynamics. There are not many studies, specially in spinal circuits that explore this in great detail. Another strength is the use of the turtle model, which allows the characterization of neuronal discharge in a relatively intact animal.

Essential revisions:

1) The manuscript requires serious editorial revisions to more clearly describe the results. This will be critical to make the text more interesting for a general readership.

– The discussion is of broad interest and compares the data obtained in the turtle spinal cord with data obtained in numerous networks. However, the Introduction is too specialized, and does not provide a general perspective. The reader is immediately confronted with the fluctuation/mean-driven regimes, without an explanation why this is interesting in a general context. It is not clear what the authors mean e.g. with networks being in unresponsive quiescence and saturation. A thorough revision of the Introduction is critical. There is a body of literature on "fluctuation-driven regimes in which excitatory and inhibitory inputs are balanced (e.g. van Vreeswijk and Sompolinsky, Science 1996), and recent in-vivo electrophysiology in cortical circuits supports this idea, e.g. Renart et al. Science 2010. The Introduction fails to make this general point, and also fails to discuss how this study is different from the more "traditional way" to study spinal circuits.

As also indicated by one of the reviewers: "The writing obscures the logic of the work".

– The same need for editorial revision applies to the Results section. Careful proof-reading is needed: see e.g. the first paragraph of the Results makes little sense; Figure 3—figure supplement 2 legend; Figure 3 legend confusing the grey and green histograms; and so on. Most seriously: An entire sub-section of the results is missing, explaining and describing Figure 4

Present the data in a logical manner.i) Establish necessary and sufficient conditions for the existence of "fluctuation-driven" regime:a) Is the subthreshold input approximately Gaussian? [Figure 5]b) Do the neurons have supralinear input-output functions in the subthreshold regime? [Figure 3]ii) How often are neurons in the fluctuation-driven regime? [Figure 6]iii) Does the mode of operation (fluctuation- vs mean-driven) depend on E/I balance? [Figure 4]iv) Can the fluctuation-driven regime be detected across the spinal motor networks? [Figure 2]v) Is it stable between animals and behaviors? [Figure 2]vi) How often and how much of the population is in the fluctuation-driven regime? [Figure 7]

2) Following this logic one notes that a key analysis step is missing. Figure 7 uses the duration of time spent in high irregularity (CV2 > CVcrit) to detect the fluctuation-driven regime, as a proxy for the direct measurements of threshold possible in the intracellular data (Figure 6). In some respects, this is their main result – that 50% of the neurons spend more than 50% of their time in this regime. To show that using CV2 can detect this regime, the authors should apply this same analysis of CV2 to their intracellular data: they need to show it is indeed a proxy for the fluctuation-driven regime.

The authors note that there is different irregularity of firing between the intracellular and extracellular recordings (Consequently, it is not possible to specifically validate the CV2 threshold chosen for the extracellular data. Nonetheless, the authors need to show that in principle this CV2 analysis can recover the fluctuation-driven regime in the intracellular data.

3) In the theoretical work on networks, the network state is considered to be stationary. This means that measures of the spiking activity such as firing rate distributions and CVs of inter-spike-intervals represent very long-time averages and, in fact, in a simulation will converge to the theoretical predictions if ever-longer time intervals are used to evaluate them. In short, there is a proper self-consistent network state for which such statistical measures can be estimated. In the case of the present work the activity is highly non-stationary. It seems the authors take advantage of a sort of separation of time scales in that the bursts, which drive the scratching behavior, as shown in Figure 1 for example, are long compared to the inter-spike interval within the burst. However, it seems this non-stationarity should introduce additional variability to any measures of spiking or subthreshold activity beyond the effects of a pure ‘fluctuation-driven' or ‘mean-driven' regime. The authors should find a way to better characterize (quantify) this, since it is one of the important messages of the paper.

The theory on fluctuation-driven versus mean-driven network states does not take into account variable or adaptive thresholds. The authors should also discuss how this might affect such states, e.g. make fluctuation-driven more robust, or less robust? Specifically, a strongly variable threshold would create the appearance of a fluctuation-driven regime, even given approximately constant input (because the timing of ISIs then depends on the noise in the threshold, not the noise in the input). To solve this, the authors could perhaps estimate the degree of non-stationarity by estimating the local variance in the threshold within the burst e.g. computing Var(threshold) in overlapping windows of 10 ISIs. In that way, they could determine if there is only strong variation at the start and end of the bursts (as suggested in some of the example figures), and so would little affect their conclusions.

4) Unfortunately, the authors have not differentiated between interneurons and motoneurons. Yet, they talk about the implications for the rhythm-generating network. Without identifying whether these neurons were either interneurons or motoneurons we don't know whether the difference in the mean/fluctuation driven neuron population is explained by the different types of neurons. The authors should discuss this caveat or even better show for a subset of interneurons (identified for example by staining) that the different types of discharge patterns are present among these neurons. From the data presented here, we don't know whether e.g. the spike distribution characterized here is only representative for motoneurons.

5) The Gaussian distribution of synaptic inputs is not directly demonstrated by actually characterizing the distribution of synaptic events, which is a caveat. The authors deduce this from the somatic membrane potential (*V_m_* = RI) which has a Gaussian distribution – and they show that it does [Figure 3 panels E-G; and Figure 3—figure supplement 2]. Given that this is such an important aspect the authors shouldn't show this key result as an inset of panel A of Figure 3—figure supplement 2. This should be a main figure to illustrate that all neurons have an approximately symmetrical distribution of membrane potentials between spikes (as the skewness is close to 0). And the caveat should be clearly stated that the authors did not perform a characterization of synaptic inputs.

6) Similarly hidden is the definition of the threshold, which is another key point. The authors define the threshold at the maximum slope of *dV_m_/V_m_* when *dV_m_* > 0 [Figure 4—figure supplement 1, panel C], but do not explain it well. The authors need to make this clearer.

Also, in Figure 4—figure supplement 1G, the authors show that the threshold increases for increasing firing rate. In Figure 4 on the other hand, the dashed lines seem to indicate that the threshold is constant during the duration of the burst, despite the fact that the firing rate is clearly changing in time. This apparent discrepancy should be addressed.

[Editors' note: further revisions were requested prior to acceptance, as described below.]

Thank you for resubmitting your work entitled "Lognormal firing rate distribution reveals prominent fluctuation-driven regime in spinal motor networks" for further consideration at *eLife*. Your revised article has been favorably evaluated by Eve Marder (Senior editor), and three reviewers, one of whom is a member of our Board of Reviewing Editors. The reviewers noted considerable improvement, in both prose and organisation of the content. In terms of the prose, the Introduction now does a much better job of laying out the issues in understanding the dynamical regimes of neuronal networks; and the Discussion nicely links the findings to a range of issues. In particular, with the new results on the physical distribution of the "fluctuating" regime neurons, the Discussion now makes clear the biggest contribution of this study: that we have to think of all neuronal networks, even spinal ones, as acting en masse to generate dynamics, not as a collection of arbitrarily labelled individuals.

The Reviewing editor notes however, there are some remaining issues that need to be addressed before acceptance, as outlined below:

1) Please consider changing the flow of the Abstract:

"When spinal circuits generate rhythmic movements it is important that the neuronal activity remains within stable bounds to avoid saturation and to preserve responsiveness. Here, we simultaneously record from hundreds of neurons in lumbar spinal circuits and establish the neuronal fraction that operates within either a ‘mean-driven’ or a ‘fluctuation-driven’ regime. Fluctuation-driven neurons have a ‘supralinear’ input-output curve, which enhances sensitivity, whereas the mean-driven regime reduces sensitivity. We find…”.

2) Introduction:

The Introduction has greatly improved! I have some minor comments, though:

2a) In "Which is generated by large circuits primarily in the spinal cord…” replace large circuits with "neuronal network". (Recent studies indicate that locomotion is generated by coupled "microcircuits" – hence the term "large circuits" is very misleading). State "in the spinal cord and medulla" too, since you refer to breathing, which is generated in the medulla.

2b) In the Introduction I would also like to see reference and discussion of the concept that different speeds of locomotion seem to be generated by different microcircuits (see studies by El Manira, Fetcho, etc.). And then differentiate your work from this concept by stating that there must also be a control of different intensities… and then go to your concept of a pool of mean-driven neurons, etc.

3) "This view was essentially predicted much earlier in random walk models [Gerstein and Mandelbrot, 1964]". Follow with a statement like: "However, this concept has been forgotten in explaining locomotion. Yet, it has been adapted to explain cortical processing". Please state something along these lines. I believe it is important to emphasize that this paper tries to apply lessons and approaches that are now commonly used in the cortex to better understand locomotion. In other words, the paper should further crystalize the novelty approach for the field of locomotion, and the opportunities by using approaches that are now commonly used for understanding neuronal circuits in areas other than the spinal cord and brainstem.

4) Small typo: third sentence down in subsection "Normally distributed synaptic input": "in fluctuation-driven regime" should be "in the fluctuation-driven regime".

5) Section "Normally distributed synaptic input": the key definition of fluctuation-driven neurons is here – you go on to relate everything else to this RMR-based definition. But, it is not clear; nor is it said anywhere how many neurons are defined as such. Clearer would be: "…and this forms the basis for selecting neurons in our analysis. An RMR close to 0.5 has fluctuation-driven spiking whereas a value close to 1 has mean-driven spiking (Figure 3—figure supplement 1). Therefore, we defined a neuron as fluctuation-driven if its RMR < 0.7; in our sample of intracellular recordings we found x/68 neurons in this regime."

6) Section "Mean *V_m_*across the population is normally distributed". Three issues here:i) You make use of the threshold here; but do not define it until a few pages later. Is this the same definition of threshold? If so, note it here.ii) "The IO-curve has approximately the same non-linearity across all neurons (Figure 3)". Figure 3 doesn't address the IO-curve; was this meant to refer to the theoretical scheme (i.e. Figure 1)?iii) Is this analysis only for the fluctuation-driven neurons, or all 68 neurons?

7) Section "Neuronal response-function in subthreshold domain is nonlinear". Again: is this analysis for only the fluctuation-driven neurons, or all 68 neurons?

8) Section "CV2 as an indicator of spiking regime". Reported here is no linear correlation between the time spent below threshold, and the mean CV2 of a neuron. This would seem to be an issue with later using CV2 to diagnose regimes in the population recording. A clear explanation of why this is not an issue would be good – I think the authors are saying that the time below threshold need have no *linear* relationship with CV2, because different neurons, all in the fluctuation-driven regime, will have different relationships between their CV2 and time below threshold (because of their specific AHP behavior, etc.). But still, Figure 2 implies there should be some relationship between CV2 and time-below-threshold.

That is, we should still be able to see the two broad classes of fluctuating/mean-driven neurons. One wonders if the issue is the same as in Figure 3—figure supplement 1: the data are not particularly suited for linear correlation. In that figure, correlating LTBT versus RMR, a linear correlation is poor because most of the data-points are clustered close to LTBT=1, and so points inside that cluster dominate the correlation, obscuring any relationship over the whole range of LTBT. So presumably LTBT vs CV2 has the same issue. Perhaps try estimating the distribution of RMR (or CV2) values per LTBT bin e.g. for the bin of, say, LTBT in [0.8 0.9], take the median RMR (or CV2) value. That way, you can correlate (or regress) the medians of the distribution per bin, estimating the relationship between the LTBT and the centre of mass of the values of RMR and CV2.

9) Section "Noisy threshold has no effect". It is stated that "not random, but rather due to a gradual inactivation of Na^+^-channels throughout the burst." How do we know this is the mechanism?

10) Section "Skewness preserved across behaviours". It is stated that "the ipsilateral behaviour had a slightly higher Gini-coefficient". In Figure 9 can only see 2 out of the 5 animals for which the ipsilateral behaviour has a higher Gini coefficient?

11) Figure 4. Two issues:

i) I think the Results text and/or Figure legend needs a simple explanation for how a firing rate output (y-axis) is derived from the ratio of two histograms (Figure 4) [as some of the details are already in the Methods]. I understand that this is essentially P(spike) transformed into a firing rate?

ii) Define more carefully what you mean by "sub threshold": Figure 4 shows that "sub threshold *V_m_*" far exceeds the threshold (star)! I think you mean that all these neurons are fluctuation-driven neurons, and so spend the majority of their time sub-threshold.

12) Discussion:

"In neuronal networks, spikes are generated in either in the mean- or the…” (omit "in").

13) "An intrinsic property, which is commonly believed to be involved in rhythm-generation, is the pacemaker property that can autonomously generate neuronal bursting in the absence of synaptic input [Brocard et al., 2010]". This paper is a good example. Since the authors also discuss their paper in the context of breathing it would be appropriate to also cite: Ramirez et al. 2011, PMID: 22654176 and Ramirez et al. 2004: Pacemaker neurons and neuronal networks: an integrative view (Curr Opin Neurobiol). The authors may also want to discuss Carroll and Ramirez 2013, a paper which discusses the role of pacemaker neurons, and discharge pattern in respiratory rhythm generation using a population approach as similarly applied here for the locomotor network.

---

## [Author Response]

*[…] Essential revisions:*

*1) The manuscript requires serious editorial revisions to more clearly describe the results. This will be critical to make the text more interesting for a general readership.*

*– The discussion is of broad interest and compares the data obtained in the turtle spinal cord with data obtained in numerous networks. However, the Introduction is too specialized, and does not provide a general perspective. The reader is immediately confronted with the fluctuation/mean-driven regimes, without an explanation why this is interesting in a general context. It is not clear what the authors mean e.g. with networks being in unresponsive quiescence and saturation. A thorough revision of the Introduction is critical. There is a body of literature on "fluctuation-driven regimes in which excitatory and inhibitory inputs are balanced (e.g. van Vreeswijk and Sompolinsky, Science 1996), and recent in-vivo electrophysiology in cortical circuits supports this idea, e.g. Renart et al. Science 2010. The Introduction fails to make this general point, and also fails to discuss how this study is different from the more "traditional way" to study spinal circuits.*

*As also indicated by one of the reviewers: "The writing obscures the logic of the work".*

The authors would like to thank for the detailed and constructive review, which has been very helpful in improving the manuscript. Thank you also for bringing attention to the need for improving the writing. We have made a thorough rewrite of both the Introduction and Results sections. The Introduction is now revised with focus on 1) reaching a general audience, 2) better describing the body of literature on balanced networks, and 3) stating how our study is different than traditional spinal investigations.

– The same need for editorial revision applies to the Results section. Careful proof-reading is needed: see e.g. the first paragraph of the Results makes little sense; Figure 3—figure supplement 2 legend; Figure 3 legend confusing the grey and green histograms; and so on. Most seriously: An entire sub-section of the results is missing, explaining and describing Figure 4

We made a mistake regarding explaining Figure 4. It is now included and revised. We have also split up several of the figures in order to improve the logic of the presentation. We have made a careful proof reading of the result sections and figure captions. The confusion of grey and green histograms is (hopefully) also cleared. We have rewritten and moved the first paragraph in the Results section to the method section as “Definition of Fluctuation and mean—driven spiking” for the interested reader. We have generally focused on making the presentation more logic and coherent with the introduction.

Present the data in a logical manner.i) Establish necessary and sufficient conditions for the existence of "fluctuation-driven" regime:

a) Is the subthreshold input approximately Gaussian? [Figure 5]

b) Do the neurons have supralinear input-output functions in the subthreshold regime? [Figure 3]

ii) How often are neurons in the fluctuation-driven regime? [Figure 6]

iii) Does the mode of operation (fluctuation- vs mean-driven) depend on E/I balance? [Figure 4]

iv) Can the fluctuation-driven regime be detected across the spinal motor networks? [Figure 2]

v) Is it stable between animals and behaviors? [Figure 2]

*vi) How often and how much of the population is in the fluctuation-driven regime? [Figure 7]*

We now present the data in the following way in specific subsections in the Results section:

1) “Mechanisms behind lognormal distribution and the fluctuation-regime”: Explanation of the issues, mechanisms, caveats and assumptions.

2) “Normally distributed synaptic input” (Establishing a Gaussian *V_m_*distribution and the arguments for suggesting a Gaussian current input) (Figure 3 first part).

3) “Mean *V_m_*across the population is normally distributed”. Here, we establish that the distribution of mean *V_m_* is also normally distributed across the population, which together with the non-linear FV curve, explains the population skewness in firing rate.

4) “Neuronal response–function in subthreshold domain is nonlinear”.

5) “Lognormal firing rate distribution in single neurons”.

6) “Time spent in regimes: intracellular data”.

7) “Transition between regimes by current injection”.

8) “Blocking inhibition causes change in regime”.

9) “CV2 as an indicator of spiking regime”. Establishing CV2 as a valid indicator for spiking regime, see discussion to point 2 below.

10) “Noisy threshold has no effect”. A section regarding the effects of noise in irregularity of spiking, see the comment below.

11) “Rich diversity in population firing rates”. Fluctuation regime across the population.

12) “Skewness preserved across behaviors”. Are the regimes stable across behaviors?

13) “Skewness in firing rate distribution is activity–dependent”. The population rate skewness is activity dependent in accordance with the two regime hypothesis.

14) “Occupancy within regimes across population and time”. The number of neurons in the two regimes is remarkably even.

15) “Cell types and spiking activity”. The spiking irregularity does not depend on where the in the dorsoventral axis that the units are located (see comments below).

Although this has become a lengthy presentation, it is relatively easy to navigate through and now more logically presented.

*2) Following this logic one notes that a key analysis step is missing. Figure 7 uses the duration of time spent in high irregularity (CV2 > CVcrit) to detect the fluctuation-driven regime, as a proxy for the direct measurements of threshold possible in the intracellular data (Figure 6). In some respects, this is their main result – that 50% of the neurons spend more than 50% of their time in this regime. To show that using CV2 can detect this regime, the authors should apply this same analysis of CV2 to their intracellular data: they need to show it is indeed a proxy for the fluctuation-driven regime.*

*The authors note that there is different irregularity of firing between the intracellular and extracellular recordings (Consequently, it is not possible to specifically validate the CV2 threshold chosen for the extracellular data. Nonetheless, the authors need to show that in principle this CV2 analysis can recover the fluctuation-driven regime in the intracellular data.*

We agree that this link between CV2 and fluctuation regime is important. We have now written a whole section about the matter in the Results section. First, we have the small sample analyses, which are already included in the manuscript. In Figure 4,Figure 6(now Figure 2,Figure 7 and Figure 8) we demonstrate a connection between CV2 and the fluctuation regime (although with some potential bias as noted). 1) CV2 and subthreshold versus supra-threshold spiking (new Figure 2) in two sample cells. 2) A sample neuron receiving depolarizing current has more spiking (smaller CV2, new Figure 7) consistent with less positive skewness in firing rate distribution (new Figure 7). When increasing the inward current by unbalancing the synaptic input, a similar decrease in irregularity is observed (new Figure 9). To substantiate the validation further, we have performed new analyses using four metrics to validate CV2 as a metric for fluctuation regime:

1) CV2 and the Return map ratio – should be anti-correlated if CV2 is a measure of the fluctuation regime.

The more fluctuations the closer the return map ratio is to 0.5. The less fluctuations in *V_m_*the closer the return map ratio is to 1. Therefore the mean CV2 sampled across the neuronal population should be anti-correlated with the mean return map ratios. We inspected the correlation between the return map ratio and the mean CV2 for all neurons that we recorded intracellularly. This is already reported in the manuscript (though hidden well) in the Figure 3—figure supplement 1 Panel E. There was a significant negative correlation (R=-0.34, P=0.005).

2) CV2 and the cumulative time below threshold of *V_m_*– should be positively correlated.

We tested for a potential relationship between CV2 and the cumulative time spent of *V_m_* below threshold for individual neurons. The threshold depends on spike rate and we used the most hyperpolarized, since it was most conservative. Nevertheless, we did not find a significant correlation between this measure and the mean CV2. Perhaps the lack of correlation is due to the variable but cell specific after-hyperpolarization, which randomly may induce a large fraction of time spent by *V_m_*below threshold even when the neuron is in mean—driven regime. This is also discussed now in the section.

3) CV2 and the skewness of Firing rate distribution should be positively correlated.

We estimated the skewness of the individual firing rate distributions for all neurons and plotted it against their mean CV2 values (see Figure 12). There was a significant positive correlation between the two, regardless of whether the firing rate distribution was plotted on a logarithmic or linear scale (R_Log = 0.43, p=0.0003, and R_Lin= 0.41, p=0.0006).

Author response image 1.**DOI:**
http://dx.doi.org/10.7554/eLife.18805.024

4) CV2 and local depolarization should be anti-correlated, if the *V_m_* is above threshold (see new Figure 8—figure supplement 1). In Figure 8—figure supplement 1 there is a lack of correlation between CV2 and *V_m_* before blocking inhibition (fluctuation regime). After removing inhibition, *V_m_* is depolarized and now there is an anti-correlation between CV2 and *V_m_*. From this we conclude that if the neuron is in the mean-driven regime the CV2 is a proxy for the depolarization above threshold. We performed a similar test of local CV2 and depolarization for all neurons (without any pharmacology). We observed a majority of cells had a negative correlation (blue histogram in Figure 13). Further, all the cells with significant correlations (red histogram, n=16/68) had anti-correlation between *V_m_*and CV2. This suggest that for the minority of neurons that where in the mean—driven regime there was a significant anti-correlation between CV2 and depolarization.

Author response image 2.**DOI:**
http://dx.doi.org/10.7554/eLife.18805.025

We have now report these analyses in the section is called “CV2 as an indicator of spiking regime”.

*3) In the theoretical work on networks, the network state is considered to be stationary. This means that measures of the spiking activity such as firing rate distributions and CVs of inter-spike-intervals represent very long-time averages and, in fact, in a simulation will converge to the theoretical predictions if ever-longer time intervals are used to evaluate them. In short, there is a proper self-consistent network state for which such statistical measures can be estimated. In the case of the present work the activity is highly non-stationary. It seems the authors take advantage of a sort of separation of time scales in that the bursts, which drive the scratching behavior, as shown in Figure 1 for example, are long compared to the inter-spike interval within the burst. However, it seems this non-stationarity should introduce additional variability to any measures of spiking or subthreshold activity beyond the effects of a pure ‘fluctuation-driven’ or ‘mean-driven’ regime. The authors should find a way to better characterize (quantify) this, since it is one of the important messages of the paper.*

Yes. We are well aware that the activity is highly non-stationary, which is the reason we chose time—dependent measures such as CV2, which helps us to demonstrate that neurons are not either in one or the other regime, but can move between them. For this reason we developed the metrics TIF and SIF, to quantify the time spent by the whole population in the fluctuation— versus mean—driven regime. These metrics do not require stationarity, but rather embrace the temporal complexity. We also find the effect of non-stationarity within the burst cycle to have only minor effect on for instance CV2 values as discussed above, when the spiking is in the fluctuation—regime. There was no correlation between the CV2 and the changes in mean *V_m_* (new Figure 8—figure supplement 1). Nevertheless, we do agree that this non-stationarity could introduce an added variability in the measures, which are important to the conclusions for our work, for instance the *V_m_*distribution and its standard deviation (although the agreement between temporal and spike-triggered estimates suggests otherwise (Figure 3)). We have clarified this assumption in the text (second section in Results section):

“The motor activity was clearly non–stationary, which means that the spike activity was likely to move between the fluctuation– and mean–regime. Nevertheless, the rhythmic activity possessed a separation of timescales in the sense that the activity between cycles (∼1 s) contained much larger excursions in *V_m_*than within cycles (∼2-400 ms). The mean *V_m_*did not change much and for practical purposes it could be considered constant within a cycle. In the following analysis of the intracellular data we regarded the dynamics in *V_m_*as stationary within a cycle – well aware that the comparison to theoretical models, which are based on assumption of stationarity, should be taken with a grain of salt.”

*The theory on fluctuation-driven versus mean-driven network states does not take into account variable or adaptive thresholds. The authors should also discuss how this might affect such states, e.g. make fluctuation-driven more robust, or less robust? Specifically, a strongly variable threshold would create the appearance of a fluctuation-driven regime, even given approximately constant input (because the timing of ISIs then depends on the noise in the threshold, not the noise in the input). To solve this, the authors could perhaps estimate the degree of non-stationarity by estimating the local variance in the threshold within the burst e.g. computing Var(threshold) in overlapping windows of 10 ISIs. In that way, they could determine if there is only strong variation at the start and end of the bursts (as suggested in some of the example figures), and so would little affect their conclusions.*

Yes, the threshold is very much adaptive. A randomly varying threshold would make the spiking look irregular even for a non-random inward current. However, the adaptation in threshold that we see is not random (as you point out). It is primarily due to increase in firing rate (new Figure 2—figure supplement 1). In this figure, there is a considerable change in threshold due to the history and length of the previous ISI. As the firing rate increases, the threshold depolarizes, which is likely due to a gradual increase in fraction of inactivated Na-channels. If the irregularity that we observe in spike times was due to a noisy variable threshold, we should see the same irregularity regardless of the depolarization, i.e. regardless of whether the neuron is in the subthreshold or supra-threshold domain and the length of the previous ISI. Nevertheless, we see a strong decrease in irregularity when the mean *V_m_*is depolarized above threshold, see e.g. Figure 1, new Figure 2, Figure 7,Figure 9(and supplement).

In order to test the intrinsic noise of the threshold, without the effect of gradual inactivation of Na^+^ channels, we looked at the first spike of a cycle over many cycles (n=51). In this way the channels had had enough time to fully de-inactive after the previous spike. In this way, when analyzing a sample cell with irregular spiking, we found the variance in the distribution of threshold for the first spike in a cycle to be small compared with the variance of the distribution of *V_m_*fluctuations. In Figure 14 we show a concatenated trace of the *V_m_*of the first spike and the 100 ms prior leading up to the spike (and 6 ms after peak). The threshold is shown as a red dot, and the black lines represent mean +/- SD (broken lines).

Author response image 3.**DOI:**
http://dx.doi.org/10.7554/eLife.18805.026

The *V_m_*seem to have larger variability than the thresholds. As it turns out, the SD of the *V_m_*was approximately four times higher than the SD of the threshold and a variance of *V_m_*, which was 17.4 times higher than that of the threshold. Distributions of the *V_m_*and first threshold are shown in Figure 15.

Author response image 4.**DOI:**
http://dx.doi.org/10.7554/eLife.18805.027

Hence, the spiking irregularity primarily comes from variability in the synaptic potentials, rather than fluctuations in threshold. We have added a section about this called “Noisy threshold has no effect” in the Results section including some discussion about the phenomenon.

Regarding whether the adaptive threshold and robustness, it will generally make the IO-function more sub-linear increase high-pass filter properties. As the input is increased the fewer spikes will be added for the same amount of input, which is a sublinear response. Hence, the adaptation will help to curb and stabilize the network activity especially in the mean driven regime. It is unclear what the effect is in the fluctuation regime, but since the general rate is lower, the adaptation will be smaller.

4) Unfortunately, the authors have not differentiated between interneurons and motoneurons. Yet, they talk about the implications for the rhythm-generating network. Without identifying whether these neurons were either interneurons or motoneurons we don't know whether the difference in the mean/fluctuation driven neuron population is explained by the different types of neurons. The authors should discuss this caveat or even better show for a subset of interneurons (identified for example by staining) that the different types of discharge patterns are present among these neurons. From the data presented here, we don't know whether e.g. the spike distribution characterized here is only representative for motoneurons.

Yes, it is indeed a caveat that we do not have more information of neurons subtypes especially whether they are interneurons (INs) versus motoneurons (MNs), as you point out. However, MNs generally represents only a small minority of the neuronal population. In the turtle, for instance, the fraction of MN to IN is 1:8, and similar values are found in other vertebrates (1:6 in chick and 1:13 in mouse, see e.g. Walloe et al., J Neurosci 2011). For this reason alone, our sample is likely to contain a large fraction of IN.

To further substantiate the implication of the fluctuation—versus mean—driven spiking for the entire population, we have performed an additional analysis of our data. The MNs are primarily located in the ventral horn, whereas INs are primarily located in the medial horn. Therefore there is a higher chance that a neuron in the ventral portions is a MN as opposed to one located in the medial portions. Thus depth of the electrode location can help infer the cellular identity.

We utilized the information of the location of our electrode array, and combined it with the spatial dependence of waveform shape of units. Using trilateration (a method to locate sources in space, like global positioning system GPS), where we assume the waveform amplitude decays with distance as 1/distance^2, we were able to determine their approximate location in the rostro-caudal and dorso-ventral axes. In the Figure 10 is shown the electrode shank lay-out (A), and the location of three sample units (color coded) on the different electrodes (B). The waveforms are the actual recordings on the electrode leads (black dots). The approximate locations of the units using trilateration are shown as red, blue and green rings.

This was repeated for all the isolated units over all shanks, probes and animals (total number of units across five animals= 226 + 249 + 190 + 58 + 198= 921). The approximate locations of all units are shown as gray dots scattered along a composite shank (C). Since the shanks span more than 200 μm we are thus able to determine the approximate location in the ventral versus medial horn. The irregularity of the spiking activity was calculated as the mean CV2 and indicated together with the depth in an additional scatter plot (D). The depths were compensated for the locations of individual shanks with respect to the spinal cord surface (note the different y-axes).

The irregularity had no discernible dependence on depth. Units were grouped according to their location (as indicated) to test for a depth-dependent distribution of irregularity in spiking (compare histograms in E). The distribution of irregular spiking across the population had no obvious dependence on the depth.

The most parsimonious interpretation of these data is that the fluctuation—driven spiking regime is both present and equally prominent in all the neurons, regardless of whether the cell body is in the ventral horn or in the medial horn.

Since the issue of cell identity has been raised by other researches in previous discussions of our work, we have decided mention this analysis in the Results section, with its own section and figure (new Figure 10).

*5) The Gaussian distribution of synaptic inputs is not directly demonstrated by actually characterizing the distribution of synaptic events, which is a caveat. The authors deduce this from the somatic membrane potential (V_m_ = RI) which has a Gaussian distribution – and they show that it does [Figure 3 panels E-G; and Figure 3—figure supplement 2]. Given that this is such an important aspect the authors shouldn't show this key result as an inset of panel A of Figure 3—figure supplement 2. This should be a main figure to illustrate that all neurons have an approximately symmetrical distribution of membrane potentials between spikes (as the skewness is close to 0). And the caveat should be clearly stated that the authors did not perform a characterization of synaptic inputs.*

We have now made the Figure 3; Figure 3—figure supplement 2 inset panel into its own figure panel (now Figure 3), including an illustration of what a skewness of +/- 1 looks like. We have also taken out the part of the instantaneous firing rate distribution in Figure 3 and made it into its own figure (now Figure 6).

Regarding the caveat, we have now explicitly written that we measure the *V_m_* distribution and that we assume it is an indication of a symmetric distribution of synaptic current. We added the following text in the Results section “Mechanisms behind lognormal distribution and the fluctuation regime”:

“The synaptic current within a cycle is difficult to assess, but rather than the mean current, we were primarily interested in the fluctuations in current, which we could approximate from *V_m_* and Ohm’s law under the following conditions. Between spiking, the mean *V_m_* was just below threshold and did not change its value much. Therefore the voltage–activated conductances were approximately constant such that there was an Ohmic relationship between synaptic current and *V_m_*. This is likely justified for neurons in fluctuation–regime, since the conductance is often high, rapid and dominated by balanced E/I synaptic input [Destexhe et al., 2003; Kumar et al., 2008]. The high conductance suppresses the coupling between *V_m_* and intrinsic conductance in a divisive manner [Kolind et al., 2012; Tiesinga et al., 2000]. Thus, in the fluctuation–regime the non–Ohmic contributions were likely smaller and the *IV_m_* -relationship more linear than in the mean–driven regime. “

And added the following text in the Results section “Normally distributed synaptic input”:

“Nevertheless, the minimal requirement for confirming the two–regime hypothesis for the single neuron is that the synaptic current (not the synaptic potentials) is Gaussian (Figure 1). As we argued earlier, if there is an Ohmic relationship between current and potential, which there likely could be during high–conductance states, then this requirement would be granted. More importantly, now that we do find a Gaussian *V_m_*–distribution, it is difficult to contemplate a non-linear *IV_m_*-relationship, which would result in such a symmetric distribution. The synaptic input current would have to have a finely matched inverse distribution to cancel out this non–linearity in order to achieve a symmetric *V_m_*–distribution. A more parsimonious explanation therefore is that, since the *V_m_*–distribution is Gaussian and the *V_m_* does not vary much, it is a result of a linear transformation of normally distributed synaptic currents.“

*6) Similarly hidden is the definition of the threshold, which is another key point. The authors define the threshold at the maximum slope of dV_m_/V_m_ when dV_m_ > 0 [Figure 6—figure supplement 1, panel C], but do not explain it well. The authors need to make this clearer.*

The definition of threshold is based on the work by Sekerli et al. (their “method II”) published in IEEE Trans Biomed Eng 2004. We have now more clearly stated the definition of spike threshold in the Materials and method section:

“We use a definition of the action potential threshold, which is based on the phase plot of *V_m_*versus the derivative *dV_m_*/dt. This is the second method reported in Sekerli et al. [2004]. The threshold is found as the point in the trajectory in phase space, where there is a strong departure from rest prior to the cycle. Since *dV_m_/dt* is proportional to the membrane current, this point represents a strong initiation of the inward current. Defining the slope of *V_m_*in time, *f = dV_m_/dt*, the threshold is defined as the largest peak in second derivative with respect to V in phase space, i.e. the maximum of d2f/dV_m_2 (red dots, Figure 2—figure supplement 1). This is the point with the largest acceleration from baseline prior to the peak of the action potential. The *V_m_* trace was low–pass filtering at 5000 Hz to reduce the vulnerable to electrical noise of the estimates of derivatives. “

*Also, in Figure 4—figure supplement1G, the authors show that the threshold increases for increasing firing rate. In Figure 6 on the other hand, the dashed lines seem to indicate that the threshold is constant during the duration of the burst, despite the fact that the firing rate is clearly changing in time. This apparent discrepancy should be addressed.*

Yes. This was meant as the lowest threshold, to make the metric “time below threshold” the most conservative measure. We have now explicitly state “the smallest threshold” everywhere in the text where it applies.

[Editors' note: further revisions were requested prior to acceptance, as described below.]

*Thank you for resubmitting your work entitled "Lognormal firing rate distribution reveals prominent fluctuation-driven regime in spinal motor networks" for further consideration at eLife. Your revised article has been favorably evaluated by Eve Marder (Senior editor), and three reviewers, one of whom is a member of our Board of Reviewing Editors. The reviewers noted considerable improvement, in both prose and organisation of the content. In terms of the prose, the Introduction now does a much better job of laying out the issues in understanding the dynamical regimes of neuronal networks; and the Discussion nicely links the findings to a range of issues. In particular, with the new results on the physical distribution of the "fluctuating" regime neurons, the Discussion now makes clear the biggest contribution of this study: that we have to think of all neuronal networks, even spinal ones, as acting en masse to generate dynamics, not as a collection of arbitrarily labelled individuals.*

*The Reviewing editor notes however, there are some remaining issues that need to be addressed before acceptance, as outlined below:*

*1) Please consider changing the flow of the Abstract:*

*"When spinal circuits generate rhythmic movements it is important that the neuronal activity remains within stable bounds to avoid saturation and to preserve responsiveness. Here, we simultaneously record from hundreds of neurons in lumbar spinal circuits and establish the neuronal fraction that operates within either a ‘mean-driven’ or a ‘fluctuation-driven’ regime. Fluctuation-driven neurons have a ‘supralinear’ input-output curve, which enhances sensitivity, whereas the mean-driven regime reduces sensitivity. We find…”.*

Yes it’s better. The Abstract now reads: “When spinal circuits generate rhythmic movements it is important that the neuronal activity remains within stable bounds to avoid saturation and to preserve responsiveness. Here, we simultaneously record from hundreds of neurons in lumbar spinal circuits and establish the neuronal fraction that operates within either a ‘mean-driven’ or a ‘fluctuation–driven’ regime. Fluctuation-driven neurons have a ‘supralinear’ input-output curve, which enhances sensitivity, whereas the mean-driven regime reduces sensitivity. We find a rich diversity of firing rates across the neuronal population as reflected in a lognormal distribution and demonstrate that half of the neurons spend at least 50% of the time in the ‘fluctuation–driven’ regime regardless of behavior. Because of the disparity in input–output properties for these two regimes, this fraction may reflect a fine trade–off between stability and sensitivity in order to maintain flexibility across behaviors.”

*2) Introduction:*

*The Introduction has greatly improved! I have some minor comments, though:*

*2a) In "Which is generated by large circuits primarily in the spinal cord…” replace large circuits with "neuronal network". (Recent studies indicate that locomotion is generated by coupled "microcircuits" – hence the term "large circuits" is very misleading). State "in the spinal cord and medulla" too, since you refer to breathing, which is generated in the medulla.*

Done.

2b) In the Introduction I would also like to see reference and discussion of the concept that different speeds of locomotion seem to be generated by different microcircuits (see studies by El Manira, Fetcho, etc.). And then differentiate your work from this concept by stating that there must also be a control of different intensities… and then go to your concept of a pool of mean-driven neurons, etc.

We have now added a sentence in the Discussion section about distinct microcircuits in zebrafish: “Distinct motor tasks have been reported to be divided among dedicated microcircuits in zebrafish [Ampatzis et al., 2014; Bagnall and McLean, 2014; Fetcho and McLean, 2010]. Nevertheless, do all neurons, which are dedicated to a particular motor activity, spike at approximately the same rate?”

*3) "This view was essentially predicted much earlier in random walk models [Gerstein and Mandelbrot, 1964]". Follow with a statement like: "However, this concept has been forgotten in explaining locomotion. Yet, it has been adapted to explain cortical processing". Please state something along these lines. I believe it is important to emphasize that this paper tries to apply lessons and approaches that are now commonly used in the cortex to better understand locomotion. In other words, the paper should further crystalize the novelty approach for the field of locomotion, and the opportunities by using approaches that are now commonly used for understanding neuronal circuits in areas other than the spinal cord and brainstem.*

We have added the following sentence: “This view was essentially predicted much earlier in random walk models [Gerstein and Mandelbrot, 1964]. The concept of balanced E/I is now an integrated part of understanding network processing in cortex and elsewhere, but for some reason it has been forgotten in understanding spinal motor networks, with the exception of a few isolated studies [Berg et al., 2007; Petersen et al., 2014].”

*4)* Small typo: third sentence down in subsection *"Normally distributed synaptic input": "in fluctuation-driven regime" should be "in the fluctuation-driven regime".*

Done.

*5) Section "Normally distributed synaptic input": the key definition of fluctuation-driven neurons is here – you go on to relate everything else to this RMR-based definition. But, it is not clear; nor is it said anywhere how many neurons are defined as such. Clearer would be: "…and this forms the basis for selecting neurons in our analysis. An RMR close to 0.5 has fluctuation-driven spiking whereas a value close to 1 has mean-driven spiking (Figure 3—figure supplement 1). Therefore, we defined a neuron as fluctuation-driven if its RMR < 0.7; in our sample of intracellular recordings we found x/68 neurons in this regime."*

We have now changed the text to the following: “An RMR close to 0.5 has fluctuation–driven spiking whereas a value close to 1 has mean–driven spiking (Figure 3—figure supplement 1). Therefore, we defined a neuron as fluctuation-driven if its RMR < 0.7; in our sample of intracellular recordings we found 50/68 neurons in this regime.”

6) Section "Mean V_m_ across the population is normally distributed". Three issues here:i) You make use of the threshold here; but do not define it until a few pages later. Is this the same definition of threshold? If so, note it here.

We have now added the following sentence: “The value used for Vthres here is the mean of the estimated thresholds for all spikes (see below).”

*ii) "The IO-curve has approximately the same non-linearity across all neurons (Figure 3)". Figure 3 doesn't address the IO-curve; was this meant to refer to the theoretical scheme (i.e. Figure 1)?*

Yes. There is a mistake in the figure reference. We have removed the figure reference, since we introduce the figure in the following section. We have now corrected the text accordingly: “If we assume, when normalizing *V_m_* this way, the IO-curve has approximately the same nonlinearity across all neurons, the population distribution of firing rates will also be skewed due to the nonlinear transformation of the normally–distributed input (Figure 3—figure supplement 2) to a lognormally–distributed output.”

*iii) Is this analysis only for the fluctuation-driven neurons, or all 68 neurons?*

Yes the analysis of the distribution of mean *V_m_* across the population was done for all 68 neurons. We now explicitly state that in the section:

“This distribution, i.e. (*V_m_* − Vthres)/σ, turns out to also be normally distributed with a mean around 3σ from threshold (Figure 3—figure supplement 2, plotted for all n = 68 neurons).”

*7) Section "Neuronal response-function in subthreshold domain is nonlinear". Again: is this analysis for only the fluctuation-driven neurons, or all 68 neurons?*

Yes, Figure 6 were done for all 68 neurons. We have now stated this explicitly in the text: “To capture the curvature we fitted both a power–law and an exponential for all n = 68 neurons and the curvature had a weak negative correlation…”.

*8) Section "CV2 as an indicator of spiking regime". Reported here is no linear correlation between the time spent below threshold, and the mean CV2 of a neuron. This would seem to be an issue with later using CV2 to diagnose regimes in the population recording. A clear explanation of why this is not an issue would be good – I think the authors are saying that the time below threshold need have no* linear *relationship with CV2, because different neurons, all in the fluctuation-driven regime, will have different relationships between their CV2 and time below threshold (because of their specific AHP behavior, etc.). But still, Figure 6 implies there should be some relationship between CV2 and time-below-threshold.*

Yes we agree. It seems that there is a bias in the way we estimate time spent below threshold as we write in the section. Perhaps one could improve the measuring technique and confirm a relationship between the two. We have not pursued the issue further. We have changed the “lack of correlation” to “lack of linear relationship” in the text.

*That is, we should still be able to see the two broad classes of fluctuating/mean-driven neurons. One wonders if the issue is the same as in Figure 3—figure supplement 1: the data are not particularly suited for linear correlation. In that figure, correlating LTBT versus RMR, a linear correlation is poor because most of the data-points are clustered close to LTBT=1, and so points inside that cluster dominate the correlation, obscuring any relationship over the whole range of LTBT. So presumably LTBT vs CV2 has the same issue. Perhaps try estimating the distribution of RMR (or CV2) values per LTBT bin e.g. for the bin of, say, LTBT in [0.8 0.9], take the median RMR (or CV2) value. That way, you can correlate (or regress) the medians of the distribution per bin, estimating the relationship between the LTBT and the centre of mass of the values of RMR and CV2.*

Yes. This type of data is for sure not so suited for linear correlation, especially since the metric is bound between 0 and 1 and as you point out, the points are clustered. We have reanalyzed the data by sectioning the RMR into bins to see if there is more of a trend, but it did not come out conclusive. We suggest leaving the issue as it is and rely on the other more solids metrics at this point.

*9) Section "Noisy threshold has no effect". It is stated that "not random, but rather due to a gradual inactivation of Na^+^-channels throughout the burst." How do we know this is the mechanism?*

Well, we do not know for sure, but there is a general consensus about the mechanism, see e.g. Henze and Buzsaki 2001, Grigonius et al. 2016, Azouz and Gray 2000, and Munoz and Fuentealba 2012. We have included the Henze and Buzsaki 2000 citation in the text, since this reference contains a detailed discussion about the matter.

*10) Section "Skewness preserved across behaviours". It is stated that "the ipsilateral behaviour had a slightly higher Gini-coefficient". In Figure 9 can only see 2 out of the 5 animals for which the ipsilateral behaviour has a higher Gini coefficient?*

Yes. We have now deleted the statement "the ipsilateral behaviour had a slightly higher Gini-coefficient”.

*11) Figure 4. Two issues: i) I think the Results text and/or Figure legend needs a simple explanation for how a firing rate output (y-axis) is derived from the ratio of two histograms (Figure 4) [as some of the details are already in the Methods]. I understand that this is essentially P(spike) transformed into a firing rate?*

Yes, we often get the question of how the firing rate relates to the histograms. We have now changed the caption slightly, to include a small explanation.

“Figure 4: Fluctuation–driven spike–response curve is supralinear. (A) The empirical probability of evoking a spike in a small window as a function of *V_m_* is determined using spike–triggered overlays. The probability distribution is estimated as the *V_m_*–distribution of trajectories prior to spike–onset (gray histogram, 1.7 ms prior to peak) normalized with the total (temporal) *V_m_*–distribution (green histogram). Dividing this probability by the sampling interval gives the firing rate (see Materials and methods).”

We also provide more detailed explanation in the Materials and methods section:

“The relationship between firing rate, ν, and membrane depolarization is based on the assumption that spikes occur as a random renewal point–process i.e. a Poisson process. The rate is directly related to the probability, P, of a spike occurring in a small time window at a certain time t:

P (t; t + ∆t) = ν∆t

The window ∆t has to be small such that the chance of getting more than one spike in the window is negligible. The firing rate can thus be defined in terms of the probability of achieving a spike in an infinitesimally small time window [Gerstner et al., 2014]:

ν(t)= limP(t;t+∆t)/∆t

This definition of ν is also called the ‘stochastic intensity’. Since the probability P is strongly dependent on the depolarization of the membrane potential, the firing rate will be similarly dependent. To determine ν as a function of *V_m_* we have to empirically determine the probability, P, for the smallest possible value of ∆t, which is the sampling interval of the intracellular recordings. To get P as a function of membrane potential*, P(V_m_*), we first empirically determine the stochastic distribution of *V_m_* prior to the spike (1.5-1.7 ms prior), which we know will cause a spike. Then we normalize this distribution with the amount of time spent at each *V_m_*-level at all time. This is the estimated probability of getting a spike, P, within a small time window ∆t for a given *V_m_*, i.e. the firing rate as a function of *V_m_*. This empirical method of relating firing rate and *V_m_* was relatively recently invented [Jahn et al., 2011] and used in determining IO properties of e.g. motoneurons [Vestergaard and Berg, 2015].”

*ii) Define more carefully what you mean by "sub threshold": Figure 4 shows that "sub threshold V_m_" far exceeds the threshold (star)! I think you mean that all these neurons are fluctuation-driven neurons, and so spend the majority of their time sub-threshold.*

Yes, we agree the idea of “sub threshold” is confusing, especially when the threshold changes. To circumvent this, we have removed the term subthreshold altogether from the figure caption: “Figure 4: Fluctuation–driven spike–response curve is supralinear. (A) The empirical probability of evoking a spike in a small window as a function of *V_m_* is determined using spike–triggered overlays. The probability distribution is estimated as the *V_m_*–distribution of trajectories prior to spike–onset (gray histogram, 1.7 ms prior to peak) normalized with the total (temporal) *V_m_*–distribution (green histogram). Dividing this probability by the sampling interval gives the firing rate (see methods).”

12) Discussion: "In neuronal networks, spikes are generated in either in the mean- or the…” (omit "in").

Done.

*13) "An intrinsic property, which is commonly believed to be involved in rhythm-generation, is the pacemaker property that can autonomously generate neuronal bursting in the absence of synaptic input [Brocard et al., 2010]". This paper is a good example. Since the authors also discuss their paper in the context of breathing it would be appropriate to also cite: Ramirez et al. 2011, PMID: 22654176 and Ramirez et al. 2004: Pacemaker neurons and neuronal networks: an integrative view (Curr Opin Neurobiol). The authors may also want to discuss Carroll and Ramirez 2013, a paper which discusses the role of pacemaker neurons, and discharge pattern in respiratory rhythm generation using a population approach as similarly applied here for the locomotor network.*

Thank you for bringing these papers to our attention. Especially Carroll and Ramierez 2013 is interesting. We have now included citation of these papers at the relevant locations in the manuscript.